# Blue-shifted ancyromonad channelrhodopsins for multiplex optogenetics

Elena G Govorunova[1], Oleg A Sineshchekov[1], Hai Li[1], Yueyang Gou[2,3], Hongmei Chen[2,3], Shuyuan Yang[4], Yumei Wang[1], Stephen Mitchell[5], Alyssa Palmateer[5], Leonid S Brown[5], François St-Pierre[3,6,7,8], Mingshan Xue[2,3,9], John L Spudich[1]*

[1]Center for Membrane Biology, Department of Biochemistry and Molecular Biology, The University of Texas Health Science Center at Houston McGovern Medical School, Houston, United States; [2]The Cain Foundation Laboratories, Jan and Dan Duncan Neurological Research Institute at Texas Children's Hospital, Houston, United States; [3]Department of Neuroscience, Baylor College of Medicine, Houston, United States; [4]Department of Chemical and Biomolecular Engineering, Rice University, Houston, United States; [5]Department of Physics and Biophysics Interdepartmental Group, University of Guelph, Guelph, Canada; [6]Department of Biochemistry and Molecular Biology, Baylor College of Medicine, Houston, United States; [7]Systems, Synthetic, and Physical Biology Program, Rice University, Houston, United States; [8]Department of Electrical and Computer Engineering, Rice University, Houston, United States; [9]Department of Molecular and Human Genetics, Baylor College of Medicine, Houston, United States

*For correspondence:
John.L.Spudich@uth.tmc.edu

## eLife Assessment

This **important** study describes newly identified light-gated ion channel homologs (channelrhodopsins, ChRs) in several protist species, with a primary focus on the biophysical characterization of ChRs of ancyromonads. The authors employed a powerful combination of bioinformatics, manual and automated patch-clamp electrophysiology, absorption spectroscopy, and flash photolysis. Additionally, they evaluated the applicability of the newly discovered anion-conducting ChRs in cortical neurons of mouse brain slices and in living *C. elegans* worms. The evidence supporting most of the claims is **compelling**, and this work will be of interest to the microbial rhodopsin community and neuro- and cardioscientists utilizing optogenetics in their research.

**Abstract** Light-gated ion channels from protists (channelrhodopsins or ChRs) are optogenetic tools widely used for controlling neurons and cardiomyocytes. Multiplex optogenetic applications require spectrally separated molecules, which are difficult to engineer without disrupting channel function. Scanning numerous sequence databases, we identified three naturally blue-shifted ChRs from ancyromonads. They form a separate branch on the phylogenetic tree and contain residue motifs characteristic of anion ChRs (ACRs). However, only two conduct chloride, whereas the closely related *Nutomonas longa* homolog generates inward cation currents in mammalian cells under physiological conditions, significantly exceeding those by previously known tools with similar spectral maxima (peak absorption at ~440 nm). Measurements of transient absorption changes and pH titration of purified proteins combined with mutant analysis revealed the roles of the residues in the photoactive site. Ancyromonad ChRs could be activated by near-infrared two-photon illumination, a

technique that enables the deeper-tissue optogenetic activation of specific neurons in three dimensions. Both ancyromonad ACRs allowed optogenetic silencing of mouse cortical neurons in brain slices. *Ancyromonas sigmoides* ACR (*Ans*ACR) expression in cholinergic neurons enabled photoinhibition of pharyngeal muscle contraction in live worms. Overall, our results deepen the mechanistic understanding of light-gated channel function and expand the optogenetic toolkit with potent, blue-shifted ChRs.

## Introduction

Channelrhodopsins (ChRs) are retinylidene proteins acting as photoreceptors that mediate photomotility in green flagellate algae (*Sineshchekov et al., 2002*) and are also found in other protist lineages. The chromophore is attached via a retinylidene Schiff base (RSB) linkage to a conserved lysine residue in the seventh transmembrane helix (TM7). Upon photoexcitation, ChRs generate passive ionic currents across the cell membrane and are used for optical control of excitable animal cells (optogenetics; *Deisseroth, 2021*; *Piatkevich and Boyden, 2023*). The seven-transmembrane (7TM) domain is sufficient for channel activity; the role of the C-terminal domain, which comprises up to half of the polypeptide chain, remains unclear. A considerable diversity within the ChR family suggests a convergent evolution of light-gated channel function (*Govorunova et al., 2022c*). ChRs form dimers (*Volkov et al., 2017*; *Li et al., 2019*) or trimers (*Tucker et al., 2022*; *Morizumi et al., 2023*), but their ionic conductance is intrinsic to individual protomers, unlike voltage- or ligand-gated channels, in which several protomers contribute to the channel pore. Anion channelrhodopsins (ACRs) generate photoinduced anion influx in mammalian cells; cation channelrhodopsins (CCRs) generate $H^+$ and $Na^+$ influx; and kalium channelrhodopsins (KCRs) generate $K^+$ efflux (*Govorunova et al., 2022a*, *Govorunova et al., 2023*). ACRs and KCRs are used for optogenetic neuronal inhibition, and CCRs are used for neuronal activation.

Increasingly popular all-optical electrophysiology, that is simultaneous perturbation and measurement of membrane potential using light-sensitive actuators and reporters, respectively, in the same genetically defined cells (*Hochbaum et al., 2014*) requires spectrally non-overlapping optogenetic tools. Even the most red-shifted ChRs retain sufficient sensitivity to blue light due to their relatively wide spectral bandwidth (*Oda et al., 2018*; *Govorunova et al., 2020*). The development of red-light-absorbing genetically encoded fluorescent biosensors for monitoring neural activity (*Sakamoto and Yokoyama, 2025*) opened up the possibility of pairing them with blue-shifted ChRs. Molecular engineering yielded several blue-shifted ChRs (*Kato et al., 2015*), but mutagenetic perturbations of the binding pocket frequently harm channel function. A complementary approach is exploring natural ChR diversity to search for molecules with desired biophysical properties optimized by evolution. Approximately ~1000 ChR sequences are currently known, but a much smaller number has been functionally characterized (*Govorunova et al., 2022c*).

Here, we identified and characterized three ChR variants from bacterivorous ancyromonad flagellates. Ancyromonads (also known as planomonads) represent a distinct major clade near the most commonly inferred root of the eukaryote tree (*Brown et al., 2018*). We conducted automated and manual patch clamp analyses of photocurrents upon expression of ancyromonad ChR cDNAs in cultured mammalian cells under one- and two-photon excitation and monitored transient light absorption changes using detergent-purified proteins. We show that two ancyromonad ChRs are anion-selective, while the third and most blue-shifted ChR conducts metal cations. We expressed ancyromonad ACRs in mouse cortical pyramidal neurons and demonstrated photoinhibition of action potentials in acute brain slices. The nematode *Caenorhabditis elegans* is an attractive model for analyzing nervous system function by optogenetic manipulation (*Bergs et al., 2022*). We used this model organism to demonstrate that a blue-shifted ancyromonad ACR enables efficient optogenetic inhibition of pharyngeal activity upon expression in the cholinergic neurons.

## Results

### Phylogeny, spectral sensitivity, and photon flux dependence

Our bioinformatic search identified ChR homologs in the ancyromonads *Ancyromonas sigmoides*, *Fabomonas tropica*, and *Nutomonas longa*; *Ancoracysta twista*, a predatory flagellate placed in the

newly established supergroup Provora (*Tikhonenkov et al., 2022*); the diatom *Odontella aurita*; and *Paraphysoderma sedebokerense*, a chytrid-like fungus from the phylum Blastocladiomycota. Phylogenetic analysis placed the ancyromonad ChRs on a separate branch of the ACR tree together with their metagenomic homologs from the TARA Oceans database (*Figure 1A*). The *A. twista* and *O. aurita* homologs clustered with known stramenopile ACRs. The *P. sedebokerense* sequence showed only a distant homology to previously known ACRs. *Figure 1—figure supplement 1* shows a protein alignment of their 7TM domains compared with *Gt*ACR1, the best-characterized ACR from the cryptophyte *Guillardia theta* (*Govorunova et al., 2015*; *Li et al., 2019*). All these sequences exhibit a non-carboxylate residue at the primary counterion position, corresponding to Asp85 in TM3 of *Halobacterium salinarum* bacteriorhodopsin (BR), marked by the red arrow in *Figure 1—figure supplement 1*, as found in all known ACRs. The second carboxylate in the photoactive site, contributed by TM7 and corresponding to Asp212 of BR, is replaced with Glu in the *N. longa* sequence, and with Gln in the *P. sedebokerense* sequence (the blue arrow in *Figure 1—figure supplement 1*).

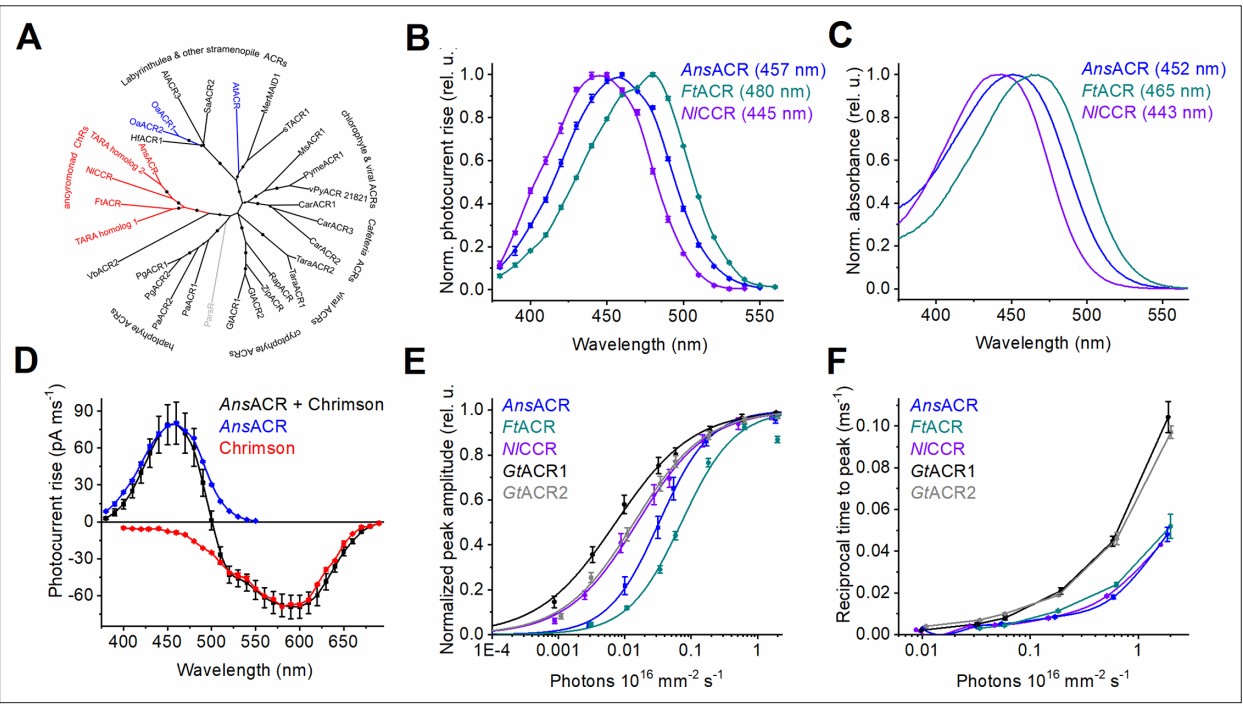

**Figure 1.** Phylogeny, spectral sensitivity, and light dependence of ancyromonad ChRs. (**A**) A maximum-likelihood phylogenetic tree of selected ChRs. The circles show bootstrap support from 40 to 100. (**B**) The photocurrent action spectra. The data points are the mean ± SEM values (n=14 cells for *Ans*ACR, and 9 cells each for *Ft*ACR and *Nl*CCR). (**C**) The absorption spectra of detergent-purified proteins. (**D**) The action spectra of photocurrents at –60 mV upon co-expression of *Ans*ACR and Chrimson, compared to each gene expressed alone. The data points are the mean ± SEM values (n=14 cells each for the co-expression and *Ans*ACR, and 4 cells for Chrimson). (**E, F**) The dependence of the peak current amplitude (E) and reciprocal time to the peak (**F**) on the photon flux density for the indicated ChRs activated at their respective $\lambda_{max}$. The data points are the mean ± SEM values (n=8 cells for each variant).

The online version of this article includes the following source data and figure supplement(s) for figure 1:

**Source data 1.** Source data for the protein names and accession numbers used to construct the tree in (A); numerical values for the data shown in (B-F).

**Figure supplement 1.** The protein alignment of the 7TM domains of ChR variants identified and characterized in this study and the previously known *Gt*ACR1.

**Figure supplement 2.** Action spectra and photocurrents of ACRs from *Ancoracysta twista* and *Odontella aurita*.

**Figure supplement 3.** Comparison of the absorption and action spectra of individual ancyromonad ChRs.

**Figure supplement 4.** Desensitization at the end of 1 s illumination.

**Figure supplement 4—source data 1.** Source data for the numerical values shown in (F).

**Figure supplement 5.** Desensitization at the end of 5-s illumination.

**Figure supplement 5—source data 1.** Source data for the numerical values shown in (F).

We expressed cDNAs encoding the 7TM domains of seven homologs fused to a C-terminal mCherry tag in human embryonic kidney (HEK293) cells and recorded photocurrents using manual patch clamping. All three ancyromonad ChRs generated robust photocurrents and showed maximal sensitivity in the blue spectral range (*Figure 1B*). Based on their ionic selectivities (see the next section), we named the *A. sigmoides, F. tropica,* and *N. longa* homologs *Ans*ACR, *Ft*ACR, and *Nl*CCR, respectively. The spectral sensitivity of the *A. twista* and *O. aurita* homologs was in the blue-green range (*Figure 1—figure supplement 2*, left). Partial replacement of Cl⁻ in the bath with non-permeable aspartate shifted reversal potential ($V_r$) to more positive values, confirming their anion selectivity (*Figure 1—figure supplement 2*, middle and right). We named them *At*ACR, *Oa*ACR1, and *Oa*ACR2. However, their photocurrents were smaller than those of the ancyromonad homologs or exhibited strong desensitization (reduction of photocurrents during illumination), so we did not characterize them in more detail. No photocurrents were detected in cells transfected with the *P. sedebokerense* homolog, although its expression and membrane targeting were evident from the tag fluorescence. Therefore, we named this protein *Pars*R, where R means 'rhodopsin'.

Next, we expressed the constructs encoding ancyromonad ChRs in *Pichia pastoris*. We purified the proteins using a mild detergent, yielding 0.25, 0.35, and 0.15 mg purified protein per L culture for *Ans*ACR, *Ft*ACR, and *Nl*CCR, respectively. The absorption spectra of the purified proteins (*Figure 1C*) were slightly blue-shifted from the respective photocurrent action spectra (*Figure 1—figure supplement 3*), likely due to the presence of non-electrogenic *cis*-retinal-bound forms. The presence of such forms, explaining the discrepancy between the absorption and the action spectra, was verified by HPLC in KCRs (*Tajima et al., 2023*; *Morizumi et al., 2023*). To test the possibility of using *Ans*ACR in multiplex optogenetics, we co-expressed it with the red-shifted CCR Chrimson (*Hochbaum et al., 2014*) fused to an EYFP tag in HEK293 cells. We measured the action spectrum of the net photocurrents with 4 mM Cl⁻ in the pipette, matching the conditions in the neuronal cytoplasm (*Doyon et al., 2016*). *Figure 1D*, black shows that the direction of photocurrents was hyperpolarizing upon illumination with $\lambda$ <500 nm and depolarizing at longer wavelengths. A shoulder near 520 nm revealed a FRET contribution from EYFP (*Govorunova et al., 2020*), which was also observed upon expression of the Chrimson construct alone (*Figure 1D*, red). *Figure 1E and F* show the dependence of the peak photocurrent amplitude and reciprocal peak time, respectively, on the photon flux density for ancyromonad ChRs and *Gt*ACRs. The current amplitude saturated earlier than the time-to-peak for all tested ChRs. *Figure 1—figure supplement 4A–E* shows normalized photocurrent traces recorded at different photon densities. Quantitation of desensitization at the end of 1-s illumination revealed a complex light dependence (*Figure 1—figure supplement 4F*). *Figure 1—figure supplement 5* shows normalized photocurrent traces recorded in response to a 5-s light pulse of the maximal available intensity and the magnitude of desensitization at its end.

## Characterization of ancyromonad ChRs by automated patch clamping

We used the fully automated planar patch clamp platform SyncroPatch 384 to characterize ancyromonad ChRs' photocurrents. This instrument uses KF-based internal and NaCl-based external solutions to promote gigaseal formation (*Supplementary file 1*). *Figure 2A–C* shows photocurrent traces evoked by 200 ms light pulses. The SyncroPatch enables unbiased estimation of the photocurrent amplitude because the cells are drawn into the wells without considering their tag fluorescence, unlike manual patch clamp studies in which the experimenter selects the fluorescent cells. *Figure 2—figure supplement 1A, B* shows that the mean *Ans*ACR photocurrent measured by the SyncroPatch was significantly larger than the mean *Ft*ACR photocurrent, and the mean *Nl*CCR photocurrent was significantly larger than that of *Platymonas subcordiformis* channelrhodopsin 2 (*Ps*ChR2), a previously known excitatory optogenetic tool with a similar blue-shifted spectrum (*Govorunova et al., 2013*; *Chen et al., 2022*).

The voltage dependencies of photocurrents (IV curves) are shown in *Figure 2D–F*. *Ans*ACR and *Ft*ACR showed similarly negative $V_r$ values, suggesting higher relative permeability to Cl⁻ than F-, as earlier found in *Gt*ACRs by manual patch clamping (*Govorunova et al., 2015*). The $V_r$ values of the peak current and that at the end of illumination were not significantly different by the two-tailed Wilcoxon signed-rank test (*Figure 2G*), indicating no change in the relative permeability during illumination. Unexpectedly, the *Nl*CCR photocurrents reversed near 0 mV (*Figure 2F and G*). One reason for this behavior could be equal permeabilities of this ChR to Cl⁻ and F-. However, when Cl⁻ in the

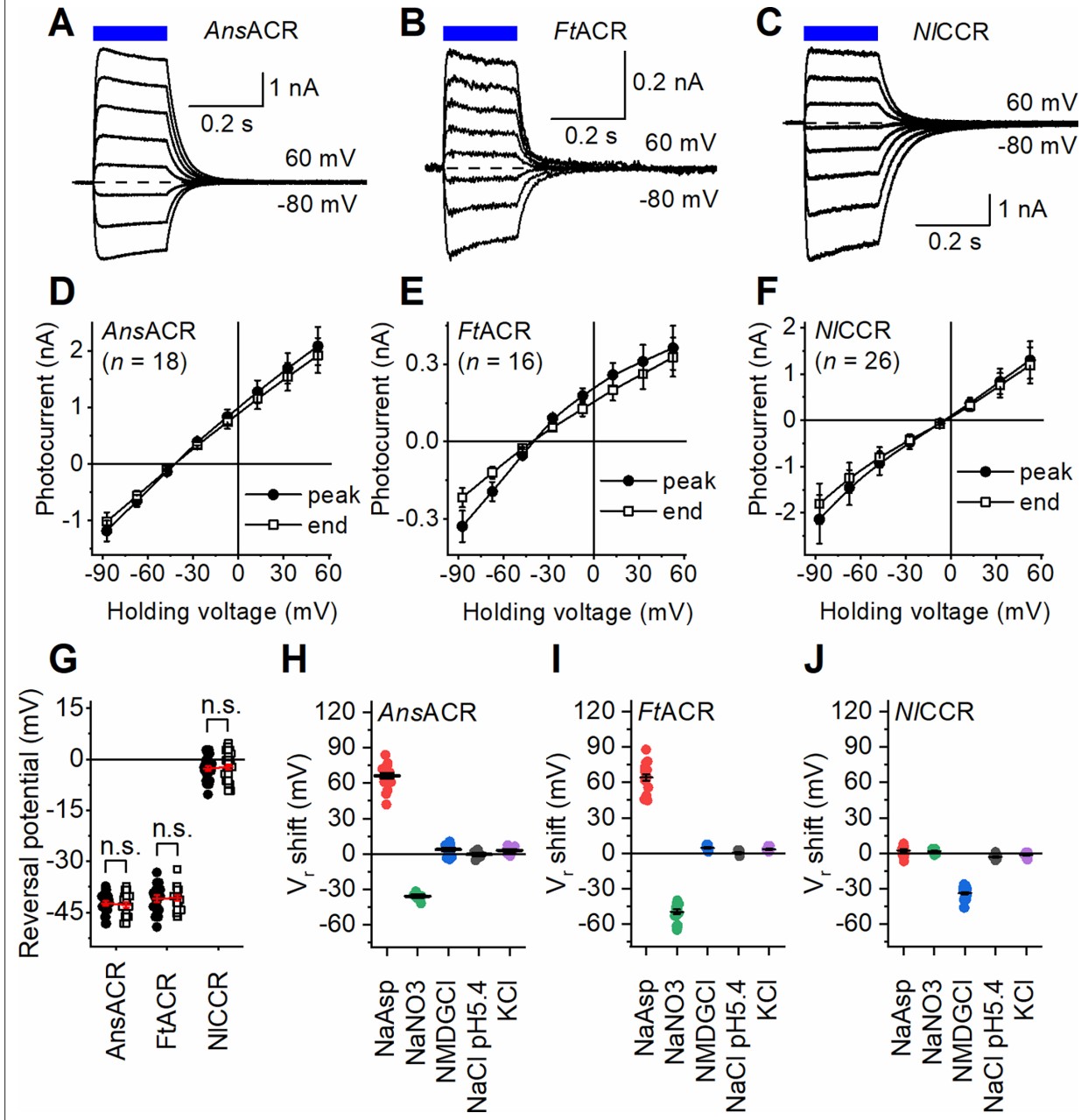

**Figure 2.** Characterization of ancyromonad ChRs by automated patch clamping. (**A–C**) Photocurrent traces recorded in response to 200 ms light pulses (470 nm) at voltages varied in 20 mV steps from −80 mV using the KF-based internal and NaCl-based external solutions. The dashed lines show the zero-current level. (**D–F**) The IV curves of the peak photocurrent (filled circles) and the current at the end of illumination (empty circles). The numbers in the parentheses are the numbers of cells sampled. (**G**) Comparison of the $V_r$ at the photocurrent peak time (filled circles) and the end of illumination (empty circles). The p-values were determined by the two-tailed Wilcoxon signed-rank test; the number of cells sampled for each variant was the same as in panels **D–F**. (**H–J**) The $V_r$ values in the indicated external solutions. The circles are the data from individual cells; the lines are the mean and SEM values.

The online version of this article includes the following source data and figure supplement(s) for figure 2:

**Source data 1.** Source data for the numbers of cells sampled and numerical values shown in (D–J).

**Figure supplement 1.** Photocurrent amplitudes of anycromonad ChRs and comparison of *Nl*CCR with *Cr*ChR2.

**Figure supplement 1—source data 1.** Source data for the numbers of cells sampled and numerical values shown in (A–D).

**Figure supplement 2.** Photocurrent traces recorded from ancyromonad ChRs by manual patch clamping.

external solution was partially replaced with bulky, non-permeable aspartate, only *Ans*ACR and *Ft*ACR showed substantial $V_r$ shifts to more positive values, which confirmed their permeability to Cl⁻, but no such shift was detected in *Nl*CCR (*Figure 2H–J*, red symbols). Control experiments conducted by manual patch clamping with the Cl⁻-based pipette solution confirmed these observations (*Figure 2— figure supplement 2*). Upon replacing Cl⁻ with $NO_3^-$, *Ans*ACR and *Ft*ACR, but not *Nl*CCR, showed $V_r$ shifts to more negative values (*Figure 2H–J*, green), as did *Gt*ACRs examined by manual patch clamping (*Govorunova et al., 2015*), which indicated higher relative permeability to $NO_3^-$ than to Cl⁻. *Ft*ACR exhibited a larger $V_r$ shift in $NO_3^-$ than *Ans*ACR (P=2.6E10-5 by the two-tailed Mann-Whitney test). Replacing Na⁺ with N-methyl-D-gluconate (NMDG⁺) resulted in a negative $V_r$ shift in *Nl*CCR but not the other tested variants (*Figure 2H–J*, blue), suggesting that *Nl*CCR is permeable to Na⁺. *Figure 2—figure supplement 1C, D* compares *Nl*CCR with the typical cation-selective *C. reinhardtii* channelrhodopsin 2 (*Cr*ChR2), assessed using the same assay. Acidifying the external solution to pH 5.4 or replacing Na⁺ with K⁺ did not affect the $V_r$ of any ancyromonad ChR (*Figure 2*, gray and violet). We conclude that only *Ans*ACR and *Ft*ACR are anion-selective, but *Nl*CCR conducts monovalent metal cations despite its sequence homology to the other two variants.

## Channel gating and photochemical transitions under single-turnover conditions

Photocurrents evoked by continuous light pulses do not accurately reflect channel kinetics because different ChR molecules absorb photons at different times. Further complications result from photon absorption by photocycle intermediates. We conducted manual patch clamp recordings upon 6-ns laser flash excitation to analyze channel gating. *Ans*ACR and *Ft*ACR photocurrents exhibited biphasic rise and decay (*Figure 3A and D*), as the earlier characterized *Gt*ACR1 (*Sineshchekov et al., 2015*). The IV curves of all kinetic components revealed the same $V_r$ values (*Figure 3—figure supplement 1A, B*), meaning no ion selectivity changes occur during the single-turnover photo- cycle. Next, we analyzed transient absorption changes in detergent-purified *Ans*ACR and *Ft*ACR. In contrast to *Gt*ACR1, we could not find a clear indication of the accumulation of a blue-shifted L intermediate. After an initial (unresolved) decay of a K-like intermediate, red-shifted absorbance temporally increased in both ancyromonad ACRs, reaching a maximum at 100–200 μs (*Figure 3B and E*, red), that is in the time domain of fast channel opening. This red-shifted intermediate could be a long-lived K or an unusual red-shifted L. In *Ft*ACR, an additional slower increase in the red-shifted absorbance likely reflected the formation of an O-like intermediate (*Figure 3E*, red). Accumulation of the M intermediate absorbing in the UV range was only 10 (*Ans*ACR) or 2 times (*Ft*ACR) slower than channel opening (*Figure 3C and F*), although in *Gt*ACR1, it was 50 times slower (*Sineshchekov et al., 2015*). Also, in contrast to prior observations in *Gt*ACR1, no temporal correlation was found between M formation and fast channel closing and between M decay and slow channel closing in ancyromonad ACRs. *Nl*CCR, the most blue-shifted among the ancyromonad ChRs we identified, has negligible absorption at 532 nm, the excitation laser's wavelength; therefore, its photochemical conversions could not be probed with our flash photolysis setup. In contrast to *Ans*ACR and *Ft*ACR, *Nl*CCR's laser-flash-induced photocurrent kinetics showed single exponential opening and closing in Cl⁻ and did not change upon its substitution with Asp⁻ in the bath solution (*Figure 3—figure supple- ment 1C and D*).

While all ACRs have a non-carboxylate residue homologous to Asp85 in BR, the second coun- terion homologous to Asp212 in BR remains protonatable (*Figure 1—figure supplement 1*, red arrow). To estimate the pKa of this counterion, we performed pH-titration of the absorption spectra (*Figure 4A–C*). The titration curves revealed two transitions in all ancyromonad ChRs. Acidification caused a transition to longer peak absorption wavelengths, as expected upon protonation of the counterion, with pK$_a$1s~3.9, 2, and 3.4 for *Ans*ACR, *Ft*ACR, and *Nl*CCR, respectively. The maximum amplitude of this transition in *Ans*ACR (~8 nm) exactly corresponded to the 8 nm red shift of the photocurrent action spectrum in the *Ans*ACR_D226N mutant (*Figure 4D*). The D226N mutation did not suppress photocurrent but simplified its kinetics (*Figure 4E*). Instead of the biphasic opening and closing observed in the WT, only two exponentials were sufficient to fit the D226N mutant's current, one for opening and one for closing. This observation suggested the role of Asp226 in channel gating, which was confirmed by analysis of the voltage dependence of the closing τ (*Figure 4F*). Upon shifting to positive voltages, both components of channel closing accelerated in the WT, but the

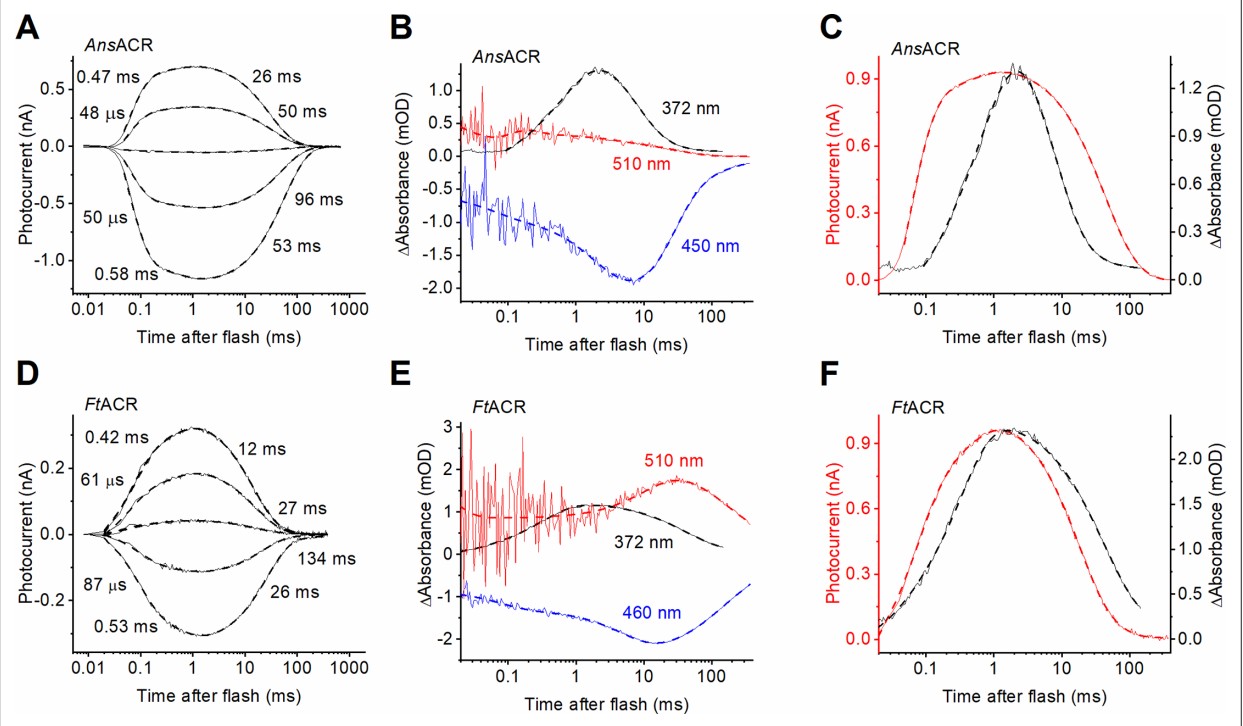

**Figure 3.** Photocurrent and transient absorption changes under single-turnover conditions in *Ans*ACR and *Ft*ACR. (**A**, **D**) Photocurrent traces of *Ans*ACR (**A**) and *Ft*ACR (**D**) evoked by 6-ns laser flashes recorded by manual patch clamping at the holding voltages increased in 30-mV steps from -60 mV. (**B**, **E**) Transient absorption changes recorded at the indicated wavelengths from detergent-purified proteins. (**C**, **F**) Comparison of the photocurrent kinetics (red, left axis) and the M intermediate kinetics (black, left axis). In all panels, the thin, solid lines are experimental recordings, and the thick, dashed lines are multiexponential approximations. The numbers are the $\tau$ values of the individual kinetic components.

The online version of this article includes the following source data and figure supplement(s) for figure 3:

**Figure supplement 1.** Current-voltage relationships of photocurrent kinetic components in the three ancyromonad ChRs and laser-flash-evoked *Nl*CCR photocurrents.

**Figure supplement 1—source data 1.** Source data for the numerical values shown in (A, B, and D).

single-exponential closing slowed in the mutant, which suggests that oppositely directed charge movements control channel closing in the WT and the mutant.

Further acidification caused a transition to shorter wavelengths with similar $pK_{a2}$s in both purified ancyromonad ACRs and only a slightly lower $pK_{a2}$ in *Nl*CCR (*Figure 4A–C*). It most probably reflects binding $Cl^-$ in the photoactive site, as in archaeal rhodopsins (*Shimono et al., 2000b*). On the other hand, in contrast to *Natronomonas pharaonis* halorhodopsin (*Váró et al., 1996*), no blue spectral shift was detected in detergent-purified *Ans*ACR at neutral pH upon an increase in the $Cl^-$ concentration (*Figure 4—figure supplement 1*), which argued against $Cl^-$ binding in the RSB region under these conditions.

Mutagenetic introduction of Glu in *Ans*ACR in the position of the primary acceptor in BR (the G86E mutation) led to the appearance of an additional spectral transition with $pK_a$ 7.4 upon pH titration of purified protein (*Figure 4G*). An extremely fast M-like UV-absorbing intermediate absent in the WT was observed in the mutant (*Figure 4H*, red). Its rise and decay $\tau$ corresponded to the rise and decay $\tau$ of the fast positive current recorded from *Ans*ACR_G86E at 0 mV and neutral pH, superimposed on the fast negative current reflecting the chromophore isomerization (*Figure 4I*, upper black trace). We interpret this positive current as an intramolecular proton transfer to the mutagenetically introduced primary acceptor (Glu86), which was suppressed by negative voltage (*Figure 4I*, lower black trace). Acidification increased the amplitude of the fast negative current ~10-fold (*Figure 4I*, black arrow) and shifted its $V_r$ ~100 mV to more depolarized values (*Figure 4—figure supplement 2A*). This can be explained by passive inward movement of the RSB proton along the large electrochemical gradient. Remarkably, the G86E mutation suppressed channel current at neutral pH, but acidification

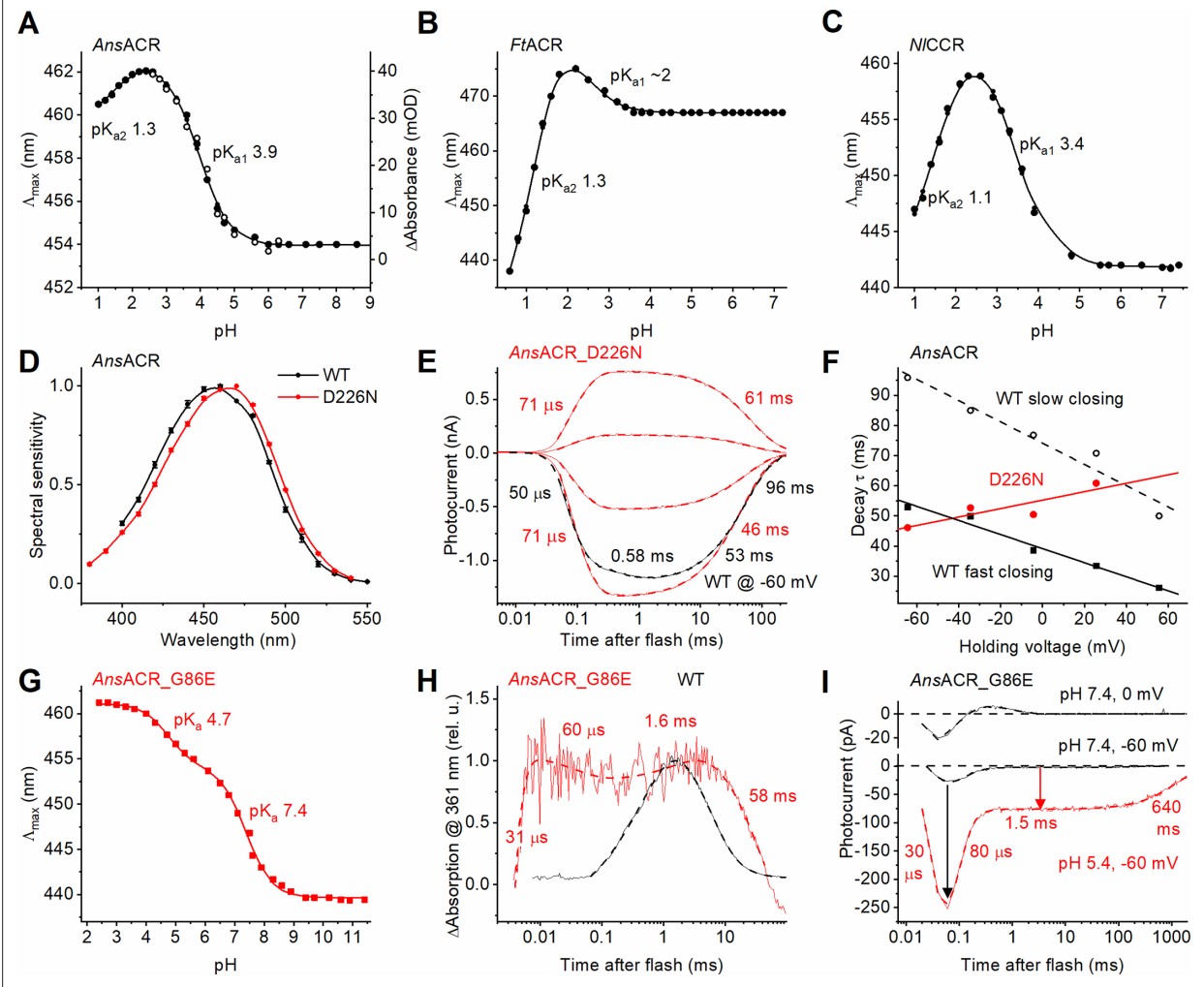

**Figure 4.** Probing the residues in the counterion positions. (**A**) pH titration of $\lambda_{max}$ (black filled circles, left axis) and maximal absorption changes (black empty circles, right axis) in wild-type *Ans*ACR. (**B**, **C**) pH titration of $\lambda_{max}$ in *Ft*ACR (**B**) and *Nl*CCR (**C**). (**D**) The photocurrent action spectrum of the *Ans*ACR_D226N mutant (red) compared to the WT (black). The data points are the mean ± SEM values (n=8 cells). (**E**) Photocurrent traces of *Ans*ACR_D226N mutant evoked by 6-ns laser flashes recorded by manual patch clamping at the holding voltages increased in 30 mV steps from −60 mV. The wild-type photocurrent trace recorded at −60 mV is shown in black for comparison. The thin lines are experimental recordings, and the thick dashed lines are multiexponential approximations. The numbers are the $\tau$ values of the individual kinetic components. (**F**) The voltage dependence of the decay components $\tau$ in the *Ans*ACR_D226N mutant (red) and the WT (black). (**G**) pH titration of $\lambda_{max}$ in *Ans*ACR_G86E mutant. (**H**) Transient absorption changes monitored at the wavelength of the M intermediate absorption in the *Ans*ACR_G86E mutant (red) compared to the WT (black). (**I**) Laser-flash-induced photocurrents of the *Ans*ACR_G86E mutant recorded at the external pH 7.4 (black) and 5.4 (red). The arrow shows the increase in the channel current upon acidification.

The online version of this article includes the following source data and figure supplement(s) for figure 4:

**Source data 1.** Source data for the numerical values shown in (A–D, F, and G).

**Figure supplement 1.** Dependence of absorption on the Cl⁻ concentration.

**Figure supplement 2.** Continued analysis of the *Ans*ACR_G86E mutant.

**Figure supplement 2—source data 1.** Source data for the numerical values shown in (A, B, and D).

**Figure supplement 3.** Photocurrents in the *Ans*ACR_Q48E mutant.

of the bath to pH 5.4 recovered it (***Figure 4I***, red arrow). The full current trace recorded under acidic conditions could be deconvoluted into four components with $\tau$ 30 µs, 80 µs, 1.5ms, and 640ms, revealing that the mutation slowed channel closing sixfold. Replacement of Cl⁻ with Asp⁻ caused an ~40 mV shift of the channel current's $V_r$ (***Figure 4—figure supplement 2B***), indicating that the mutant

channel remained Cl⁻ selective. These results confirm that the absence of a negative charge at the site corresponding to BR's primary acceptor is the ultimate condition for anion channel function.

Strong alkalization caused simultaneous depletion of absorption in the visible range, the appearance of the M-like states (at 368 nm in the wild-type *Ans*ACR and 355 nm in the *Ans*ACR_G86E mutant), and a substantial absorption increase at 297 nm, reflecting deprotonation of the RSB and a strong perturbation of the protein band (*Figure 4—figure supplement 2C*). Analysis of the pH dependence of three parameters (absorption depletion in the visible range, absorption rise in the M-like states' range, and absorption rise at 297 nm) yielded similar $pK_a$ values, which were 11.1 and 10.7 for the WT and G86E, respectively (*Figure 4—figure supplement 2D*). These high $pK_a$ values may explain the high photostability of this protein, as hundreds of laser flashes did not cause its measurable bleaching.

The glutamate in the middle of TM2 corresponding to Glu68 of *Gt*ACR1 is conserved in most ACRs, including *Ft*ACR, but is replaced with Gln in *Ans*ACR (*Figure 1—figure supplement 1*, black arrow). Introducing the Q48E mutation in *Ans*ACR accelerated channel closing and slowed channel opening, making the photocurrent rise and decay monophasic (*Figure 4—figure supplement 3A and B*).

## The retinal-binding pocket and color tuning

The $\lambda_{max}$ of rhodopsins is regulated by the retinal chromophore geometry and steric and electrostatic interactions of the chromophore with amino acid residues of the retinal-binding pocket (*Hoffmann et al., 2006*; *Karasuyama et al., 2018*). Surprisingly, *Nl*CCR, the most blue-shifted among ancyromonad ChRs, features three residues typical of red-shifted microbial rhodopsins (*Figure 5A*). The first is Phe at the primary counterion position (Asp85 in BR), also found in RubyACRs from Labyrinthulea, the most red-shifted ChRs so far identified (*Govorunova et al., 2020*). The residues homologous to BR's Met118 near the β-ionone ring and Ala215 preceding the RSB lysine in the polypeptide chain are responsible for red-shifted absorption in many microbial rhodopsins (*Shimono et al., 2000a*; *Engqvist et al., 2015*; *Oda et al., 2018*; *Oppermann et al., 2024*) but are conserved in all three blue-absorbing ancyromonad ChRs. In *Gt*ACR1 ($\lambda_{max}$ 515 nm; *Govorunova et al., 2015*; *Sineshchekov et al., 2016*), the only ACR with published atomic structures (*Kim et al., 2018*; *Li et al., 2019*), the corresponding residues are Ser97, Cys133, and Cys237 (*Figure 5B*). As expected, the S97F, C133M, and C237A mutations red-shifted the *Gt*ACR1 spectrum (*Figure 5C*). The opposite F104S, M143C, and A242C mutations at the corresponding sites in the RubyACR from *Hondaea fermentalgiana* (*Hf*ACR1) caused large blue spectral shifts (*Figure 5D*). However, the corresponding mutations F85S, M141C, and A236C red-shifted the *Nl*CCR spectrum (*Figure 5E*). The same paradoxical behavior was observed upon mutation of the Met118 homolog to Val, which blue-shifted the *Hf*ACR1 spectrum but red-shifted the ancyromonad ChR spectra (*Figure 5F–H*). Replacement of the Ala215 homolog with Cys or Ser did not change the *Ans*ACR and *Ft*ACR spectra (*Figure 5—figure supplement 1A, B*).

In blue-shifted ChRs such as *Ps*ChR2 (*Govorunova et al., 2013*) and *Klebsormidium nitens* channelrhodopsin (*Kn*ChR) (*Tashiro et al., 2021*), the position of BR's Met118 is occupied by Gly or Ala (*Figure 5A*), which, together with the Ala four residues downstream (BR's Gly122), rotates the β-ionone ring out of the plane of the polyene chain, shrinking the *p*-conjugation and blue-shifting the spectrum (*Wang et al., 2025*). The Ala corresponding to BR's Gly122 is also found in *Ans*ACR and *Nl*CCR (*Figure 5A*), but the *Ans*ACR_M134A/G and *Nl*CCR_M141A/G mutations did not change the spectra, nor did the *Ft*ACR_M140G_V144A mutation (*Figure 5—figure supplement 1A–C*). These observations suggest that the residue geometry and/or interactions in the retinal-binding pocket in ancyromonad ChRs differ from the earlier studied microbial rhodopsins.

The residue position corresponding to BR's Leu93 is the color switch between blue- and green-absorbing proteorhodopsins (BPRs and GPRs; *Man et al., 2003*). *Ans*ACR exhibits a Gln residue in this position, as do BPRs, but *Ft*ACR has Leu, as do GPRs, and *Nl*CCR has Met (*Figure 5A*). The *Ft*ACR_L96Q and *Nl*CCR_M93Q mutations blue-shifted the spectra 20 and 7 nm, respectively (*Figure 5I and J*), indicating that this residue position contributes to color tuning in ancyromonad ACRs. *Nl*CCR, the most blue-shifted among ancyromonad ChRs, differs from *Ans*ACR and *Ft*ACR at the positions corresponding to Ser89 and Glu233 (*Nl*CCR numbering), and from *Ft*ACR, also at the position of Pro235 (the corresponding residues in *Gt*ACR1 are Thr101, Asp234, and Leu236, *Figure 5A*). The S89T, E233D, and P235I mutations red-shifted the *Nl*CCR spectrum (*Figure 5K*), indicating that these three positions contribute to the blue shift of wild-type *Nl*CCR compared with *Ans*ACR and *Ft*ACR.

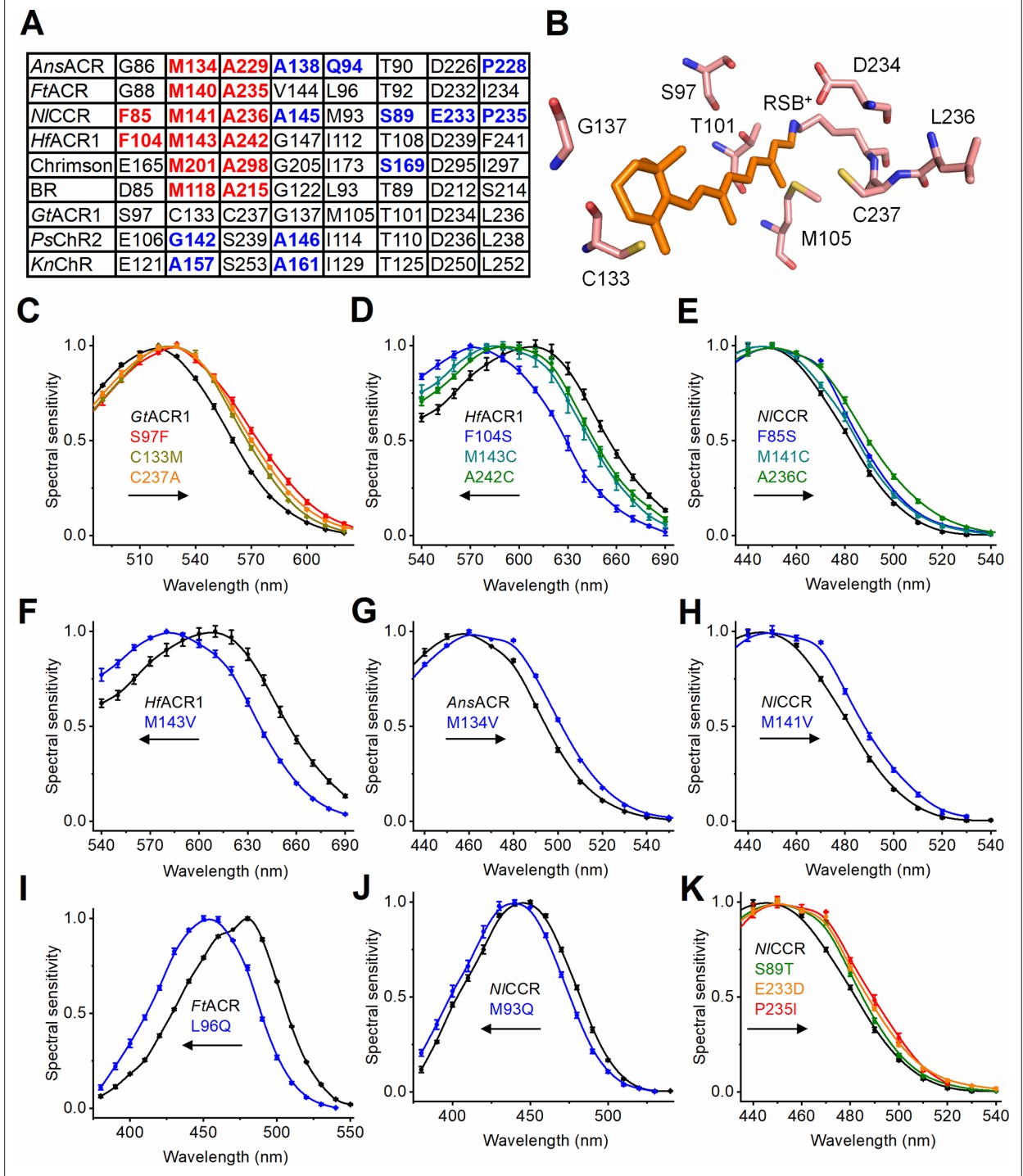

**Figure 5.** Color tuning of ancyromonad ChRs. (**A**) Amino acid residues of the retinal-binding pocket tested by mutagenesis in this study. (**B**) The corresponding residues in the *Gt*ACR1 structure (6edq). (**C–K**) The photocurrent action spectra of the indicated mutants compared to the respective WTs. The data points are the mean ± SEM values (the n values are provided in *Figure 5—source data 1*).

The online version of this article includes the following source data and figure supplement(s) for figure 5:

**Source data 1.** Source data for the numbers of cells sampled and numerical values shown in (C–K).

**Figure supplement 1.** Mutations blue-shifting other microbial rhodopsin spectra do not affect ancyromonad ChRs.

**Figure supplement 1—source data 1.** Source data for the numbers of cells sampled and numerical values shown in (A–D).

However, the T90S mutation did not change the *Ans*ACR spectrum, and the D226E mutation and a combination of the two mutations caused red spectral shifts in *Ans*ACR (*Figure 5—figure supplement 1D*). *Supplementary file 2* contains the $\lambda$ values of the half-maximal amplitude of the long-wavelength slope of the spectrum, which can be estimated more accurately from the action spectra than the $\lambda$ of the maximum.

## Two-photon excitation

Optical manipulation of neuronal activity in dense tissue commonly relies on two-photon (2P) excitation, which is based on the nearly simultaneous absorption of two infrared photons, equivalent to the absorption of one photon in the visible range (*Emiliani et al., 2022*). To determine the 2P activation range of *Ans*ACR, *Ft*ACR, and *Nl*CCR, we conducted raster scanning using a conventional 2P laser, varying the excitation wavelength between 800 and 1080 nm (*Figure 6—figure supplement 1*). All three ChRs generated detectable photocurrents with action spectra showing maximal responses at ~925 nm for *Ans*ACR, 945 nm for *Ft*ACR, and 890 nm for *Nl*CCR (*Figure 6A*). These wavelengths fall within the excitation range of common Ti:Sapphire lasers, which are widely used in neuroscience laboratories and can be tuned between ~700 nm and 1020–1300 nm. To assess desensitization, cells expressing *Ans*ACR, *Ft*ACR, or *Nl*CCR were illuminated at the respective peak wavelength of each ChR at 15 mW for 5 s. *Gt*ACR1 and *Gt*ACR2, previously used in 2P experiments (*Forli et al., 2018*; *Mardinly et al., 2018*), were included for comparison. The normalized photocurrent traces recorded under these conditions are shown in *Figure 6B–F*. The absolute amplitudes of 2P photocurrents at the peak time and at the end of illumination are shown in *Figure 6G and H*, respectively. All five tested variants exhibited comparable levels of desensitization at the end of illumination (*Figure 6I*).

## Optogenetic inhibition of cortical neurons in mouse brain slices

To test the silencing efficiencies of *Ans*ACR and *Ft*ACR in mouse brain slices, we selectively expressed their 7TM domains fused with EYFP in the layer 2/3 pyramidal neurons of the somatosensory cortex by in utero electroporation at embryonic day 15. We prepared acute brain slices from 4- to 6-week-old mice, and EYFP fluorescence was observed in the layer 1, layer 2/3, and layer 5, indicating clear *Ans*ACR-EYFP and *Ft*ACR-EYFP expression in dendrites, somata, and axons, respectively (*Figure 7—figure supplement 1A*). We performed whole-cell current clamp recordings from *Ans*ACR- or *Ft*ACR-expressing neurons (for solution compositions, see Materials and methods). *Ans*ACR- and *Ft*ACR-expressing neurons showed the resting membrane potential, input resistance, and capacitance (*Figure 7—figure supplement 1B*) similar to the typical values of untransfected cortical neurons (*Xue et al., 2014*; *Chen et al., 2020*). When positive currents were injected into the somata to excite neurons, photoactivation of *Ans*ACR and *Ft*ACR suppressed the current-evoked action potentials, demonstrating the potency of these proteins as optogenetic silencers of mouse cortical neurons (*Figure 7*). We also observed that at rest or when a small negative current (e.g. –0.1 nA) was injected, the neurons could generate a single action potential at the beginning of photostimulation (*Figure 7A and B*), possibly caused by axonal depolarization, as reported in *Gt*ACR-expressing neurons (*Mahn et al., 2018*; *Messier et al., 2018*).

Earlier studies have shown that photoactivation of *Gt*ACRs induces axonal depolarization and synaptic transmission in some ACR⁺ neurons owing to the high intracellular Cl⁻ concentration at the axons (*Mahn et al., 2018*; *Messier et al., 2018*). Indeed, when we recorded from ACR⁻ neurons in the electroporated cortical region, we found that photoactivation of either *Ans*ACR- or *Ft*ACR-induced excitatory post-synaptic currents (*Figure 7—figure supplement 1C, D*), similar to other tested light-gated Cl⁻ channels.

## Optogenetic inhibition of pharyngeal function in live *C. elegans*

To test *Ans*ACR as an optogenetic inhibitory tool in the context of an intact behaving animal, we expressed the encoding construct fused to a C-terminal EYFP tag in the *C. elegans* cholinergic neurons using the *unc-17* promoter (a scheme of the expression construct is shown in *Figure 8A*). *C. elegans* feeds on bacteria by rhythmic contractions and relaxations (pumping) of its pharynx. The cholinergic pharyngeal neurons, primarily MC neurons, entrain the pharyngeal muscle rhythm (*Trojanowski et al., 2016*). Neuronal and muscular electrical activity leading to pharyngeal contractions can be monitored non-invasively by electropharyngeogram (EPG) recording (*Raizen and Avery, 1994*). An EPG

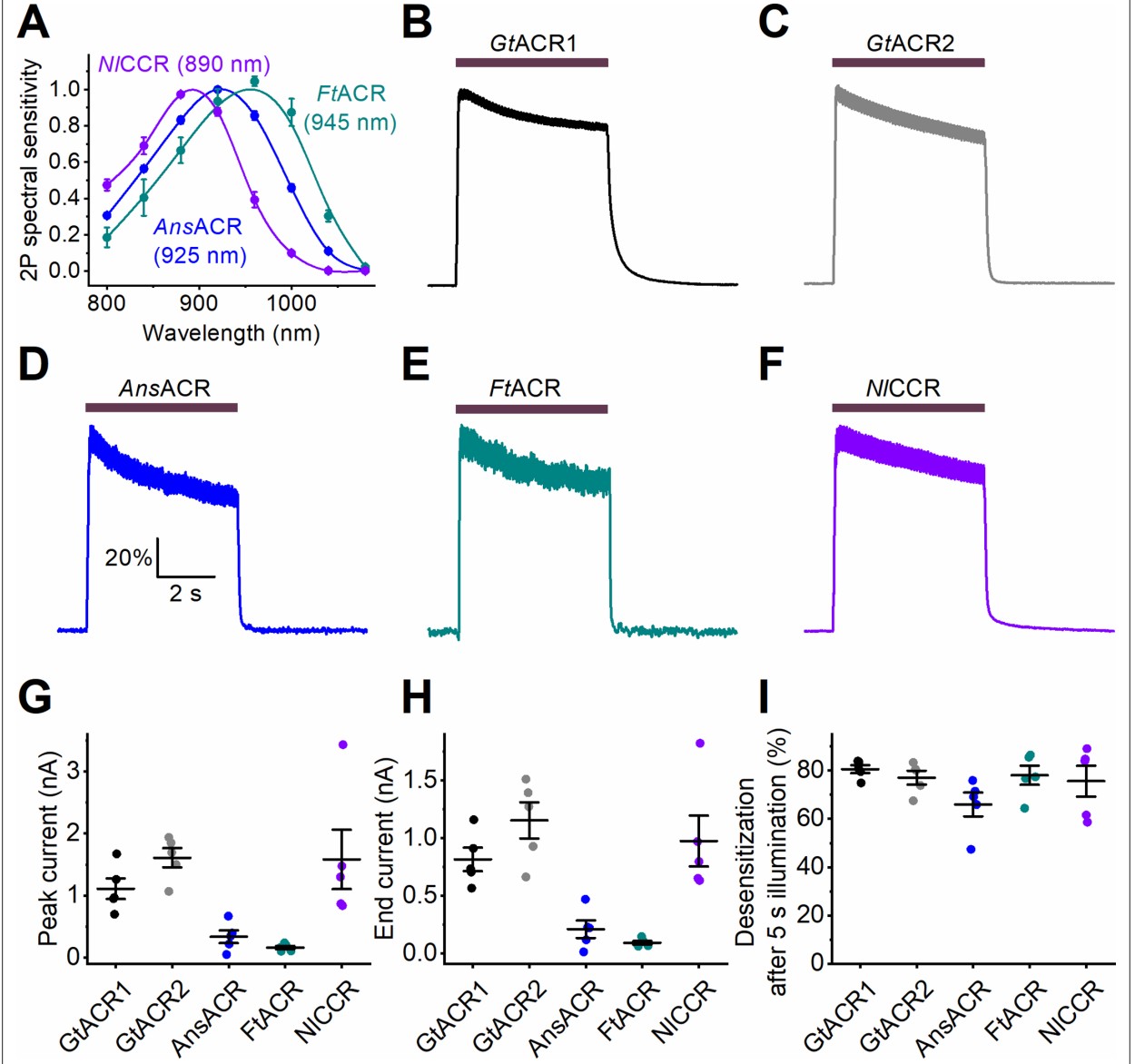

**Figure 6.** 2P excitation of ancyromonad ACRs. (**A**) The 2P photocurrent action spectra. The data points are the mean ± SEM values (n=5 cells for each variant). (**B–F**) The mean normalized photocurrent traces recorded upon 2P excitation from the indicated ChR variants (*Gt*ACR1 and *Gt*ACR2 are included for comparison) at +20 mV in the Cl⁻-based external solution (n=6 cells for each variant). The illumination (the duration of which is shown as the bars on top) was 15 mW at the $\lambda_{max}$ for each variant. (**G, H**) The amplitude of photocurrent measured at the peak time (**G**) and at the end of 5 s illumination (**H**). (**I**) Desensitization at the end of illumination. In **G–I**, the symbols are the data from individual cells, the lines are mean ± SEM values (n=6 cells for each variant). For more detail, see Materials and methods.

The online version of this article includes the following source data and figure supplement(s) for figure 6:

**Source data 1.** Source data for the numerical values shown in (A and G–I).

**Figure supplement 1.** Power dependence upon 2P illumination.

**Figure supplement 1—source data 1.** Source data for the numbers of cells sampled and numerical values shown in (B, D).

contains transients reflecting pharyngeal muscle action potentials (*Figure 8B*), the frequency of which can be easily quantified. In the presence of 10 mM serotonin required to maintain regular pharyngeal pumping, its frequency in the dark was not significantly different in the transgene and wild-type worms (4.07 ± 0.05 and 4.19 ± 0.08 Hz, respectively, mean ± SEM, n=26 transgenic and 11 wild-type worms, respectively; the p-value by the two-tailed Mann-Whitney test is 0.21), indicating that *Ans*ACR expression did not affect the pharyngeal function in the darkness. *Figure 8C* shows representative

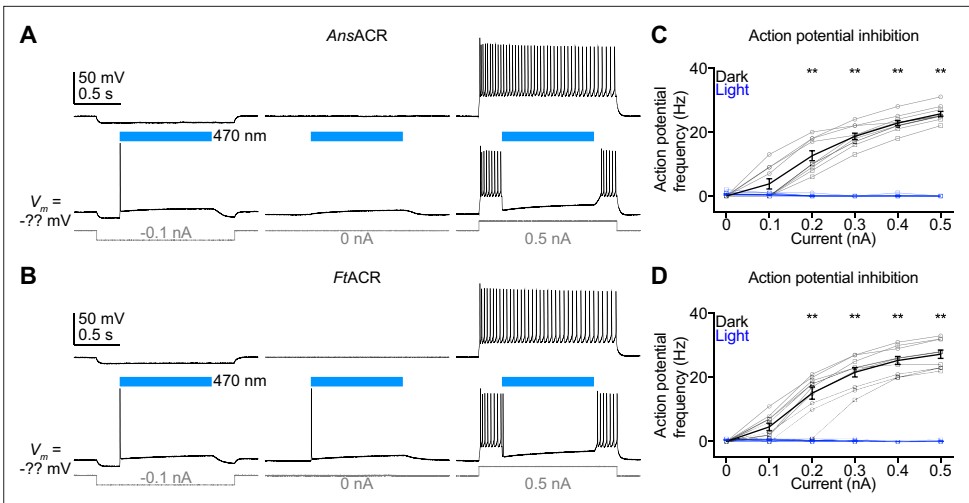

**Figure 7.** Photoactivation of *Ans*ACR and *Ft*ACR inhibits the action potentials of mouse cortical neurons. (**A**, **B**) Representative membrane voltage traces of neurons expressing *Ans*ACR (**A**) and *Ft*ACR (**B**) in response to –0.1 nA (left), 0 nA (middle), and 0.5 nA (right) injections without (top) and with (bottom) 470 nm light pulses (power density of 38.7 mW mm$^{-2}$). (**C**, **D**) The frequencies of action potentials evoked by different current injections with (blue) and without (black) photoactivation of *Ans*ACR (**C**) and *Ft*ACR (**D**). For all panels, data points from male mice are indicated by squares and female mice by circles. One male and one female mouse were used for each of the *Ans*ACR and *Ft*ACR experiments. Data are mean ± SEM. **, p≤0.01 for comparison between dark and light stimulation at 0.2–0.5 nA current injection by the multiple Wilcoxon matched-pairs signed rank test with Benjamini, Krieger, and Yekutieli's corrections.

The online version of this article includes the following source data and figure supplement(s) for figure 7:

**Source data 1.** Source data for the numerical values shown in (C, D).

**Figure supplement 1.** Characterization of *Ans*ACR and *Ft*ACR expression in cortical neurons and axonal excitatory effect.

**Figure supplement 1—source data 1.** Source data for the numbers of cells sampled and numerical values shown in (B, D).

EPG recordings from a transgenic worm fed on bacteria supplemented with all-*trans*-retinal. The onset of 470 nm illumination caused an immediate inhibition of pumping in such transgene worms but not in the WT worms fed on the same bacteria or transgene worms in the absence of retinal (**Figure 8D**). The *C. elegans* genome encodes LITE-1 and GUR-3 UV/blue light receptors (unrelated to ChRs) responsible for photoinhibition of pharyngeal pumping at high light levels (**Bhatla et al., 2015**). However, no photoinhibition was detected in the absence of retinal in either wild-type or transgenic worms. This indicates that the irradiance used in our experiments was insufficient to stimulate these endogenous photoreceptors. The magnitude of the *Ans*ACR-mediated photoinhibition depended on the irradiance (**Figure 8E**). The maximal irradiance (2.1 mW mm$^{-2}$) completely abolished the pumping for 15 s in all tested worms (n=13). In five of 13 worms, individual action potentials were observed during the second half of the 30 s illumination period, an indication of adaptation. Two independently created transgenic lines showed the same degree of photoinhibition (**Figure 8E**, filled and empty symbols). The photoinhibition was fully reversible: after switching off the maximal-irradiance light, the pumping frequency returned to the pre-illumination level with $\tau$~9 s.

## Discussion

ChRs, also known as '*Chlamydomonas* sensory rhodopsins', were first discovered as the photoreceptors guiding phototaxis and the photophobic response in the chlorophyte *C. reinhardtii* (**Sineshchekov et al., 2002**; **Govorunova et al., 2004**). Since then, ChRs have been identified in the genomes and transcriptomes of several other eukaryotic supergroups, including cryptophytes, haptophytes, stramenopiles, and alveolates (**Govorunova et al., 2015**; **Govorunova et al., 2020**; **Govorunova et al., 2021**). Furthermore, ChRs appear in the genomes of giant viruses, which likely facilitate the spread

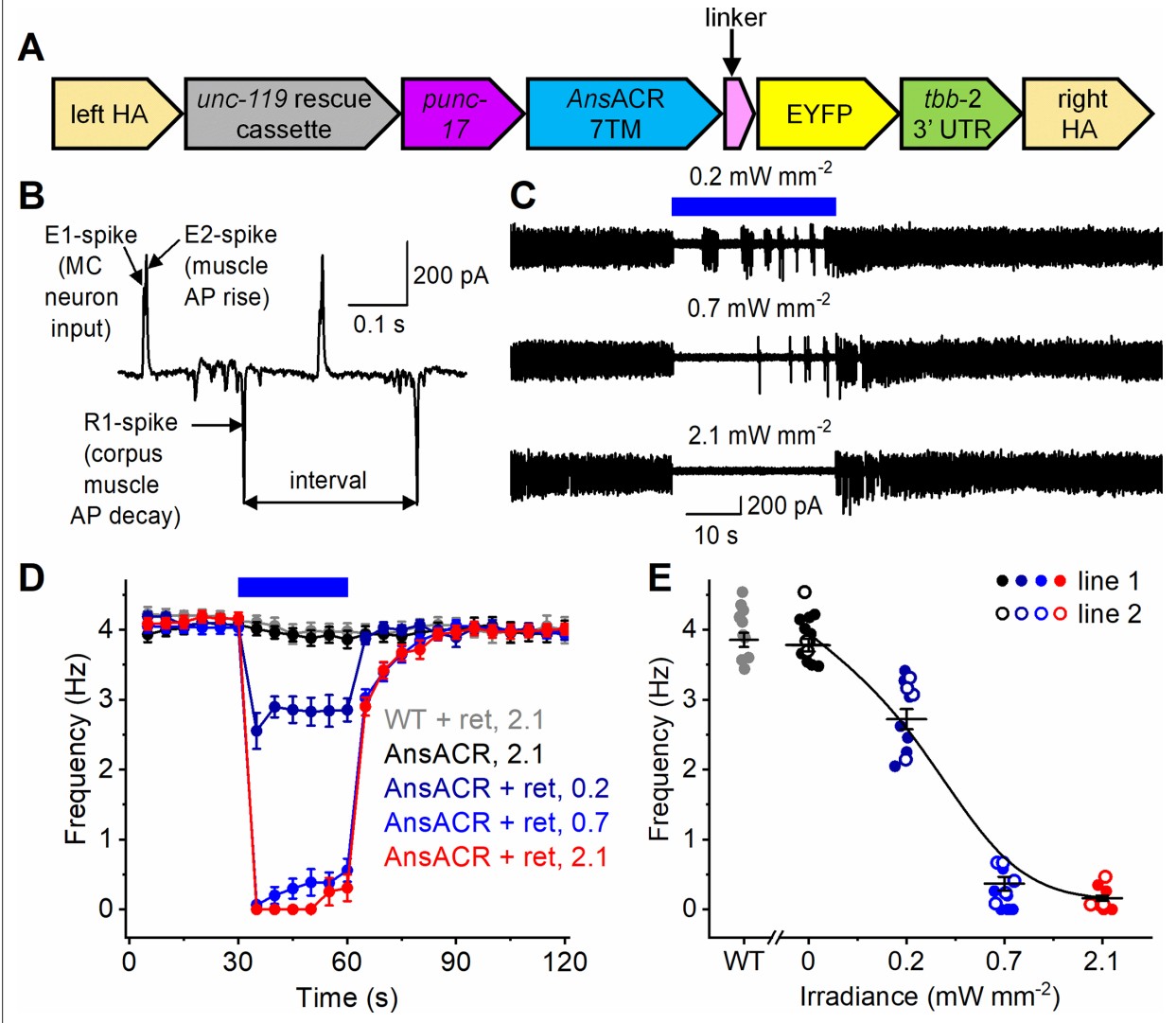

**Figure 8.** Photoinhibition of pharyngeal pumping in live *C. elegans* expressing *Ans*ACR in the cholinergic neurons. (**A**) A scheme of the genetic construct for *Ans*ACR expression in the cholinergic neurons. HA, homology arms. (**B**) A zoomed-in section of an EPG recording. AP, action potential; TB, terminal bulb. The double-headed arrow shows the interval between two successive R1 spikes used to calculate pharyngeal pumping frequency. (**C**) Electropharyngeogram recordings from an *Ans*ACR-expressing worm illuminated with 470 nm light at the indicated irradiances. The blue bar shows the duration of illumination. (**D**) The frequency of the pharyngeal pumping calculated from recordings as shown in A. The symbols are the mean values, and the error bars are the SEM values (n=11 worms for the WT and 13 worms per condition for the transgenic worms). The numbers are the irradiance values in mW mm$^{-2}$; ret is retinal. (**E**) The dependence of the pharyngeal pumping frequency on the irradiance calculated from the 30–60 s segment of the data shown in B. The symbols are the data from individual worms; the lines are the mean and SEM values. The empty and filled symbols for the transgenic worms show the data from two independently created transgenic lines.

The online version of this article includes the following source data for figure 8:

**Source data 1.** Source data for the numerical values shown in (D, E).

of ChR genes by horizontal transfer (*Rozenberg et al., 2020*; *Zabelskii et al., 2020*). Our identification and characterization of ChRs in ancyromonads, phylogenetically placed near the most commonly inferred root of the eukaryote tree (*Brown et al., 2018*), suggests that eukaryotes acquired ChR genes at the early steps of their evolution. Consistent with the role of the encoded proteins as phototaxis receptors as shown in *C. reinhardtii*, ChR genes or transcripts have been found only in protists that develop flagella at some stage of their life cycle. The diatom *O. aurita*, in which we identified ChRs, is no exception: the flagella are lost in the vegetative state of this protist but are still present in its male gametes (*Nanjappa et al., 2017*).

Prediction of biophysical properties such as ionic selectivity from protein sequences is a major unresolved problem in ChRs research. The *Nl*CCR sequence shows ~29% identity and ~49% similarity in the 7TM domain to each of the two ancyromonad ACRs and contains a neutral residue in the counterion position (Asp85 in BR), typical of all ACRs (*Figure 1—figure supplement 1*, red arrow). Yet, *Nl*CCR does not conduct anions, showing instead permeability to Na$^+$. In the earlier known ChRs, the presence of conserved Glu residues in TM2 and the TM2-TM3 loop, corresponding to Glu82, Glu83, Glu90, and Glu101 of *Cr*ChR2, correlates with cation selectivity (*Govorunova et al., 2021*). However, TM2 of *Nl*CCR contains no carboxylated residues (*Figure 1—figure supplement 1*), which suggests a unique mechanism of cation selection in this channel. *Nl*CCR is the most blue-shifted among ancyromonad ChRs and generates larger photocurrents than the earlier known *Ps*ChR2 with a similar absorption maximum (*Govorunova et al., 2013*), which makes *Nl*CCR a good candidate for optogenetic stimulation of neuronal activity with blue light.

All ancyromonad ChRs absorb light in the blue spectral range. The $\lambda_{max}$ of retinylidene proteins is determined by the energy gap between the electronic ground (S0) and first excited (S1) state of the chromophore and depends on the chromophore geometry, the protonation state of the Schiff base counterion, and the interaction of the chromophore with other residues of the retinal-binding pocket (*Ernst et al., 2014*; *Engqvist et al., 2015*; *Kato et al., 2015*). Paradoxically, the retinal-binding pockets of all three ancyromonad ChRs contain the residues corresponding to Met118 and Ala215 of bacteriorhodopsin, the well-known 'color switches' characteristic of red-shifted microbial rhodopsins. The role of the near-ring Met118 homolog in red-shifting the spectrum has been experimentally verified in *Haloquadratum walsbyi* BR (*Sudo et al., 2013*), *Gloeobacter violaceus* rhodopsin (*Engqvist et al., 2015*), archaeorhodopsin-3 (*Kato et al., 2015*), and Chrimson (*Oda et al., 2018*). It is thought that the bulky Met side chain pushes away the C7 atom of retinal, increasing the ring-chain coplanarity, expanding the π-conjugation, and red-shifting absorbance, as theoretically predicted in the *Gt*ACR1_C133M mutant (*Tsujimura et al., 2021*). The red-shifting effect of the Ala215 homolog has been demonstrated in *N. pharaonis* sensory rhodopsin II (*Shimono et al., 2000a*), *H. salinarum* BR (*Spudich et al., 2012*), Chrimson (*Oda et al., 2018*), sodium-pumping rhodopsin KR2 (*Inoue et al., 2019*), and *Mantoniella squamata* ACR1 (*Oppermann et al., 2024*), and is explained by electrostatic interactions between the polar residue in this position and the RSB. However, in ancyromonad ChRs, mutations of the Met118 homolog to smaller residues and the Ala215 homolog to polar residues caused a red spectral shift or no shift. Furthermore, the mutagenetic introduction of the ring-rotating residues responsible for the blue-shifted spectra of *Hyphochytrium catenoides* kalium channelrhodopsin 2 (*Hc*KCR2) (*Tajima et al., 2023*) and *Kn*ChR (*Wang et al., 2025*) did not change the ancyromonad ChR spectra, which suggests that either the torsion around the C6-C7 bond is already enforced by a different geometry of the Met118 homolog or that the blue-shifted absorbance of ancyromonad ChRs arises by a different mechanism. Atomic structures of ancyromonad ChRs are needed to investigate the unexpected spectral shifts we observed when mutating residues of the retinal binding pocket.

ACRs are widely used to inhibit neuronal activity with light. We evaluated *Ans*ACR and *Ft*ACR as neuronal silencers in mouse brain slices and *Ans*ACR in the context of a live animal, the nematode *C. elegans*. We previously showed that *Gt*ACRs could inhibit action potentials at the soma while triggering synaptic transmission due to high axonal Cl$^-$ reversal potential (*Messier et al., 2018*). *Ans*ACR and *Ft*ACR showed similar phenomena in our brain slice experiments. These ACRs can inhibit action potentials in cortical neurons but depolarize axonal terminals and trigger synaptic transmission at the onset of light stimulation. Fusing these new ACRs with somatodendritic trafficking motifs to reduce axonal expression (*Mahn et al., 2018*; *Messier et al., 2018*) could lead to potent inhibition with reduced axonal excitation. Nevertheless, this undesired excitatory effect needs to be taken into consideration when using ACRs.

Optogenetic inhibition of *C. elegans* pharyngeal pumping has been demonstrated earlier upon expression of the *Leptosphaeria maculans* proton-pumping rhodopsin known as Mac (*Trojanowski et al., 2014*) or *N. pharaonis* halorhodopsin (*Np*HR) (*Schüler et al., 2015*) in the cholinergic neurons. However, ion-pumping rhodopsins such as these transport only one ion per absorbed photon and, therefore, require almost 20 times higher irradiance for photoinhibition than the maximal irradiance used in this study. All ACRs, including *Ans*ACR that we tested in the worms, transport multiple anions during the open state and, therefore, are more efficient optogenetic silencers than the ion-pumping

rhodopsins. One possible explanation of the partial recovery of pharyngeal pumping that we observed after 15 s illumination, even at the highest tested irradiance, is continued attenuation of photocurrent during prolonged illumination (desensitization). However, the rate of *Ans*ACR desensitization (*Figure 1—figure supplement 4A* and *Figure 1—figure supplement 5A*) is much faster than the rate of the pumping recovery, reducing the likelihood that desensitization is driving this phenomenon. Another possible reason for the observed adaptation is an increase in the cytoplasmic Cl⁻ concentration owing to *Ans*ACR activity and hence a breakdown of the Cl⁻ gradient on the neuronal membrane. The *C. elegans* pharynx is innervated by 20 neurons, 10 of which are cholinergic (*Pereira et al., 2015*). A pair of MC neurons is the most important for regulation of pharyngeal pumping, but other pharyngeal cholinergic neurons, including I1, M2, and M4, also play a role (*Trojanowski et al., 2014*). Moreover, the pharyngeal muscles generate autonomous contractions in the presence of acetylcholine tonically released from the pharyngeal neurons (*Trojanowski et al., 2016*). Given this complexity, further elucidation of pharyngeal pumping adaptation mechanisms is beyond the scope of this study.

In summary, our characterization of ancyromonad channelrhodopsins (ChRs) reveals that their blue-shifted spectral sensitivity and unique ionic selectivity arise from distinct residue motifs not found in previously characterized ChRs. These findings broaden our understanding of how protein sequence modulates light-gated channel function. The blue-shifted absorption properties of ancyromonad ChRs hold promise for multiplexed applications alongside red-shifted indicators and warrant further evaluation across diverse experimental systems.

# Materials and methods

## Key resources table

| Reagent type (species) or resource | Designation | Source or reference | Identifiers | Additional information |
|---|---|---|---|---|
| Gene (*Ancyromonas sigmoides*) | *Ans*ACR | GenBank | PQ657777 | Encodes anion-selective ChR |
| Gene (*Fabomonas tropica*) | *Ft*ACR | GenBank | PQ657778 | Encodes anion-selective ChR |
| Gene (*Nutomonas longa*) | *Nl*CCR | GenBank | PQ657779 | Encodes cation-selective ChR |
| Gene (*Guillardia theta*) | *Gt*ACR1 | GenBank | KP171708 | Encodes anion-selective ChR |
| Gene (*Guillardia theta*) | *Gt*ACR2 | GenBank | KP171709 | Encodes anion-selective ChR |
| Recombinant DNA reagent | *Ans*ACR_pcDNA3.1 (plasmid) | This study | Addgene #232598 | PcDNA3.1 backbone |
| Recombinant DNA reagent | *Ft*ACR_pcDNA3.1 (plasmid) | This study | Addgene #232599 | PcDNA3.1 backbone |
| Recombinant DNA reagent | *Nl*CCR_pcDNA3.1 (plasmid) | This study | Addgene #232600 | PcDNA3.1 backbone |
| Recombinant DNA reagent | pAAV-CAG-*Ans*ACR-EYFP (plasmid) | This study | Addgene #238347 | pAAV-CAG backbone |
| Recombinant DNA reagent | pAAV-CAG- *Ft*ACR-EYFP (plasmid) | This study | Addgene #238348 | pAAV-CAG backbone |
| Strain, strain background (*Escherichia coli*) | DH5α | Thermo Fisher Scientific | CMC0016 | Competent cells for gene cloning |
| Strain, strain background (*Pichia pastoris*) | SMD1168 | Thermo Fisher Scientific | C17500 | Used for production of recombinant ChRs |
| Cell line (*Homo sapiens*) | HEK293 | ATCC | CRL-1573 | Used for 1P excitation patch clamp experiments |
| Cell line (*Homo sapiens*) | HEK293A | Invitrogen | R70507 | Used for 2P excitation patch clamp experiments |
| Strain, strain background (*Mus musculus*), female | ICR (CD-1) | BCM Center for Comparative Medicine | ICR (CD-1) | Used for brain slice recordings |
| Strain, strain background (*Mus musculus*), male | C57BL/6 J | Jackson Laboratory | JAX #000664 | Used for brain slice recordings |
| Genetic reagent (*Caenorhabditis elegans*) | COP2831 | This study | [pNU3704 ([uncp-17::AnsACR::EYFP::tbb-2u, unc-119(+))] II; unc-119(ed3) III | Transgenic line expressing *Ans*ACR in cholinergic neurons |
| Genetic reagent (*Caenorhabditis elegans*) | COP2832 | This study | [pNU3704 ([uncp-17::AnsACR::EYFP::tbb-2u, unc-119(+))] II; unc-119(ed3) III | Transgenic line expressing *Ans*ACR in cholinergic neurons |

*Continued on next page*

*Continued*

| Reagent type (species) or resource | Designation | Source or reference | Identifiers | Additional information |
|---|---|---|---|---|
| Commercial assay or kit | QuikChange XL | Agilent Technologies | #200516 | Site-directed mutagenesis kit |
| Commercial assay or kit | Lipofectamine LTX with Plus Reagent | Thermo Fisher Scientific | #15338100 | Used for HEK293 cell transfection |
| Commercial assay or kit | FuGENE HD transfection reagent | Promega | #E2311 | Used for HEK293A cell transfection |
| Chemical compound, drug | All-*trans*-retinal | Millipore-Sigma | #116-31-4 | Chromophore for ChRs, added after transfection |
| Chemical compound, drug | Zeocin | Thermo Fisher Scientific | #R25001 | Used for selection of transformant *P. pastoris* clones |
| Software, algorithm | MegAlign Pro | DNASTAR Lasergene | 17.1.1 | Used for sequence alignment |
| Software, algorithm | IQ-TREE | Los Alamos National Laboratory | 2.1.2 | Used for phylogeny analysis |
| Software, algorithm | iTOL | EMBL | 7 | Used for phylogenetic tree visualization |
| Software, algorithm | PyMOL | Schrödinger | 2.4.1 | Molecular visualization software |
| Software, algorithm | pClamp | Molecular Devices | 10.7 | Used for data acquisition and analysis in patch clamp experiments |
| Software, algorithm | Origin Pro | OriginLab Corporation | 2016 | Used for analysis and visualization of patch clamp data |
| Software, algorithm | Logpro | Zenodo | Logpro | Used for logarithmic noise reduction in photocurrent traces |

## Bioinformatics and molecular biology

The ChR homologs from *Ancyromonas sigmoides* strain B-70 (CCAP1958/3), *Fabomonas tropica* strain NYK3C, *Nutomonas longa* strain CCAP 1958/5 (*Torruella et al., 2015*; *Brown et al., 2018*), and *Ancoracysta twista* strain TD-1 (*Janouškovec et al., 2017*) were identified in the EukProt V3 database (*Richter et al., 2022*) using Sequnceserver BLASTP (*Priyam et al., 2019*). The *A. sigmoides*, *F. tropica*, and *A. twista* ChR sequences are also available from Dr. Andrey Rozenberg's ChR database (*Rozenberg, 2024*). The metagenomic homolog 1 was found by Sequenceserver BLASTP in the TARAeuCatV2 database (*Sunagawa et al., 2015*) accessed at the KAUST Metagenomic Analysis Platform (KMAP; *Alam et al., 2021*). The metagenomic homolog 2 was found using the search mode of BLASTP in the MATOU database (Marine Atlas of Tara Oceans Unigene plus metaG eukaryotes) (*Villar et al., 2018*) with the query sequence of *Nl*CCR. The *Odontella aurita* strain CCMP816 homologs GHBW01284417 and GHBW01118808 were identified by TBLASTN in the National Center of Biological Information (NCBI) transcriptome shotgun assembly (TSA) project GHBW00000000. The *Paraphysoderma* homolog (*Pars*R) was found in the *P. sedebokerense* strain JEL821 v. 1.0 genome assembly (*Amses et al., 2022*) by the annotation text search using bacteriorhodopsin as a keyword at the Mycocosm portal (*Ahrendt et al., 2023*).

The protein alignment was created using the MUSCLE algorithm with default parameters implemented in MegAlign Pro software v. 17.1.1 (DNASTAR Lasergene, Madison, WI) and truncated after the end of TM7. Phylogeny was analyzed with IQ-TREE v. 2.1.2 (*Minh et al., 2020*) using automatic model selection and ultrafast bootstrap approximation (1000 replicates) (*Hoang et al., 2018*). The best tree was visualized and annotated using iTOL v. 7 (*Letunic and Bork, 2024*). PyMol (v. 2.4.1, Schrödinger) was used for molecular visualization.

For expression in human embryonic kidney (HEK293) cells, mammalian codon-optimized polynucleotides encoding amino acid residues 1–265 of the *A. sigmoides* homolog, 1–268 of the *F. tropica* homolog, 1–272 of the *N. longa* coding homolog, 1–243 of the *A. twista* homolog, 1–241 of the GHBW01284417 *O. aurita* homolog, 1–242 of the GHBW01118808 *O. aurita* homolog, and 1–336 of the *P. sedebokerense* homolog were synthesized, fused to a C-terminal mCherry tag, and cloned into the pcDNA3.1(+) vector (Invitrogen, Cat. #V19520) at GenScript. Mammalian codon-optimized

polynucleotides encoding amino acid residues 1–295 of *Gt*ACR1 (Genbank Acc. #KP171708), 1–291 of *Gt*ACR2 (Genbank Acc. #KP171709), and 1–350 of *Chlamydomonas noctigama* ChR1 known as Chrimson (Genbank Acc. #KF992060) were fused to a C-terminal EYFP (enhanced yellow fluorescent protein) tag and cloned into the same vector backbone. For expression in *Pichia*, the constructs were fused with the C-terminal 8His-tag and cloned in the pPICZalpha-A vector (Invitrogen, Cat. #V19520). A QuikChange XL site-directed mutagenesis kit (Agilent Technologies, Cat. #200516) was used to introduce point mutations. For expression in the mouse cortical neurons, *Ans*ACR and *Ft*ACR were tagged with EYFP at the C-terminus and cloned into the pAAV-CAG vector.

## HEK293 cell culture and transfection

No cell lines from the list of known misidentified cell lines maintained by the International Cell Line Authentication Committee or non-human cell lines were used in this study. HEK293 cells used in one-photon (1 P) excitation experiments were obtained from the American Type Culture Collection (ATCC, Cat. #CRL-1573), authenticated by short tandem repeats (STR) profiling at ATCC, and tested negative for mycoplasma contamination by PCR analysis. The cells were plated on 2 cm diameter plastic dishes 48–72 hrs before experiments, grown for 24 hr, and transfected with 10 µl of Lipofectamine LTX with Plus Reagent (Thermo Fisher, Cat. #15338100) using 3 µg DNA per dish for manual patch clamping, and 6 µg DNA per dish for automated patch clamping. All-*trans*-retinal (Millipore-Sigma, Cat. #116-31-4) was added immediately after transfection at the final concentration of 5 µM.

For 2P excitation experiments, HEK293A cells (Invitrogen, Cat. # R70507) were authenticated by STR profiling at the M.D. Anderson Cancer Center Cytogenetics and Cell Authentication Core (CCAC) tested negative for mycoplasma contamination by PCR analysis and plated on 30–70 kDa poly-*d*-lysine-coated 12 mm circular coverslips (Carolina cover glass #0, Cat. #633009) in 24-well plates (P24-1.5H-N, Cellvis) at 30% confluence, transfected with 1.2 µL FuGENE HD transfection reagent (Promega, Cat. #E2311) using 200 ng DNA per well 48–72 hr before measurements and supplemented with all-*trans*-retinal as described above.

## Automated whole-cell patch clamp recording from HEK293 cells

Automated patch clamp recording was conducted at room temperature (21°C) with a SyncroPatch 384 (Nanion Technologies) based on a Biomek i5 automated liquid handler (Beckman Coulter), using NPC-384T S-type chips (Nanion, Cat. #222101) with one hole per well, as described earlier (*Govorunova et al., 2022b*). Transfected cells (48–72 hr after transfection) were dissociated using TrypLE Express, diluted with CHO-S-SFM-II medium (both from Thermo Fisher, Cat.# 12604013 and 31033020, respectively), and resuspended in External Physiological solution (Nation, Cat.# 08 3001). The compositions of this and other solutions used in automated patch clamp recordings and the corresponding liquid junction potential (LJP) values calculated using the ClampEx LJP calculator are listed in *Supplementary file 1*. The voltages in all IV curves for HEK293 cells studied under 1 P excitation in this manuscript were corrected for LJPs; the holding voltage values in the figures showing traces correspond to the amplifier output before the LJP subtraction. Illumination was provided with LUXEON Z Color Line light-emitting diodes (LEDs) Cat.# LXZ1-PB01 (470±10 nm) arranged in a 6×16 matrix. The forward LED current was 900 mA (which corresponded to the irradiance of ~2 mW mm$^{-2}$), the illumination duration was 200ms (limited by the LED duty cycle), and the interval between successive light pulses was 60 s. The LEDs were driven by a derivative of CardioExcyte 96 SOL (Nanion, Cat. #191003) and controlled by Biomek commands. PatchControl384 v. 2.3.0 (Nanion Technologies) software was used for data acquisition at a 5 kHz sampling rate (200 µs per point). The photocurrent amplitudes at the peak and the end of illumination were calculated using DataControl384 software v. 2.3.0 (Nanion Technologies). Further analysis was performed using the Origin Pro 2016 software (OriginLab Corporation).

## Manual patch clamp recording using 1P excitation in HEK293 cells

Manual patch clamp recordings were performed with an Axopatch 200B amplifier (Molecular Devices). The pipette solution contained (in mM) KCl 130, MgCl2 2, HEPES 10 pH 7.4, and the bath solution contained (in mM) NaCl 130, CaCl$_2$ 2, MgCl$_2$ 2, glucose 10, HEPES 10 pH 7.4. In experiments to test the relative permeability of ChRs for Cl$^-$, NaCl in the bath was replaced with Na aspartate, and in experiments in cells co-transfected with *Ans*ACR and Chrimson, KCl in the pipette solution was

replaced with K gluconate. The low-pass filter of the amplifier output was set to 2 kHz. The signals were digitized with a Digidata 1440 A (Molecular Devices) at a 250 kHz sampling rate (4 µs per point) in experiments with laser flashes, and at a 5 kHz sampling rate (200 µs per point) in experiments with continuous light pulses using pClamp 10.7. Patch pipettes with 2–3 MΩ resistances were fabricated from borosilicate glass. Laser excitation was provided by a Minilite Nd:YAG laser (532 nm, pulse width 6 ns, energy 5 mJ; Continuum). The current traces were logarithmically filtered using Logpro software (*Spudich, 2022*). Curve fitting was performed using Origin Pro software. Continuous light pulses were provided by a Polychrome V light source (T.I.L.L. Photonics GMBH) in combination with a mechanical shutter (Uniblitz Model LS6, Vincent Associates; half-opening time 0.5ms). The action spectra of photocurrents were constructed by calculating the initial slope of photocurrent recorded in response to 15 ms light pulses at the intensity <25 µW mm$^{-2}$, corrected for the quantum density measured at each wavelength, and normalized to the maximal value. To analyze the dependence of photocurrents on the photon flux density, 1 s light pulses were applied with 60 s dark intervals starting from the lowest density. Calibrated neutral density filters (Newport, Cat. #FSQ-OD50, #FSQ-OD100, #FSQ-OD150, and #FSQ-OD200) were used to adjust the density.

## Manual patch clamp recording using 2P excitation in HEK293A cells

2P excitation of *Ans*ACR, *Ft*ACR, *Nl*CCR, *Gt*ACR1, and *Gt*ACR2 expressed in HEK293A cells was conducted using an inverted microscope with multiphoton capability (A1R-MP, Nikon Instruments) at room temperature (23°C). A coverslip seeded with the transfected cells was placed in a custom glass-bottom chamber based on Chamlide EC (Live Cell Instrument) with a glass bottom made with a 24×24 mm cover glass #1 (Erie Scientific, Cat. #89082–270). Cells were perfused continuously with the external solution as described in the 1 P excitation section. Whole-cell voltage-clamp recordings were performed using a MultiClamp 700B amplifier (Molecular Devices). Cells were held at −20 mV for power dependency and spectral measurements, and at +20 mV for desensitization measurements. The holding voltages were compensated for the 4.4 mV LJP calculated using the ClampEx v.11.1 (Molecular Devices) built-in calculator. The signals were digitized with an Axon Digidata 1550B1 Low Noise system with a HumSilencer (Molecular Devices), and the current was recorded at 10 kHz using pClamp. The near-IR excitation was generated by a titanium:sapphire femtosecond laser (Chameleon Ultra II, Coherent) with a repetition rate of 80 MHz and a tuning range between 680 and 1,080 nm. Laser pulses were not pre-compensated for dispersion in the microscope optical path. Laser power was tuned using an acousto-optic modulator and delivered to the sample plane through a 40×0.95-numerical aperture (NA) objective (CFI Plan Apochromat Lambda, Nikon Instruments). Scanning across 40.96×40.96 µm regions-of-interest was achieved using resonant scanning at 33.3 Hz.

To determine the 2P action spectra of *Ans*ACR, *Ft*ACR, and *Nl*CCR, the excitation wavelength was varied from 800 to 1080 nm in 40 nm increments. At each wavelength, the laser power at the sample plane was adjusted to 7.5 mW for *Ans*ACR, 5 mW for *Ft*ACR, and 3 mW for *Nl*CCR, as measured using a microscope slide power sensor (S170C, Thorlabs). Excitation power levels were selected to elicit robust photocurrents while minimizing ChR desensitization, by operating at or near the quadratic regime, where doubling the excitation power results in an approximate fourfold increase in the initial photocurrent slope. Deviations from the target power level were kept below 10% and corrected by considering the quadratic dependence of photocurrents on power under 2P excitation. Illumination consisted of 30 consecutive raster scans (total duration ~1 s) over a 40.96×40.96 µm region (512x512 pixels), approximating the average size of a HEK293A cell. Scans were performed back-to-back without temporal gaps, except for the brief interval required for the laser to return to the starting scan position. To mitigate desensitization, we spaced illumination pulses ~54 s apart. We verified that the power ramp and spectral scan protocols caused a less than 20% reduction in the peak photocurrent, as measured at the beginning and end of each protocol using peak-wavelength light pulses at 7.5 mW for *Ans*ACR, 5 mW for *Ft*ACR, and 3 mW for *Nl*CCR. The 2P action spectra were constructed by measuring the initial linear slope of the photocurrent rise at each wavelength and plotted using Origin Pro 2016 software.

For desensitization experiments, cells expressing *Ans*ACR, *Ft*ACR, *Nl*CCR, *Gt*ACR1, and *Gt*ACR2 were illuminated near their respective peak wavelengths: *Ans*ACR (920 nm), *Ft*ACR (960 nm), *Nl*CCR (920 nm), *Gt*ACR1 (1040 nm), and *Gt*ACR2 (940 nm) at 15 mW for 5 s. Current traces were low-pass filtered at 100 Hz and, for presentation purposes, downsampled by substituting an average value for

each 100 data points. The *Ft*ACR trace was additionally smoothed by the 5-point Savitzky-Golay algorithm in Origin. Peak currents were quantified as the maximal currents over the 5 s traces. End currents were calculated as the mean current over the final 0.1 s of the 5 s photoactivation pulse.

## Expression and purification of ancyromonad ACRs from *Pichia pastoris*

The plasmids carrying the expression constructs were linearized with Sac I and delivered into the *P. pastoris* strain SMD1168 (Thermo Fisher, Cat. # C17500) by electroporation. A single colony resistant to 0.25 mg/ml zeocin (Thermo Fisher, Cat. #R25001) was picked and inoculated into buffered complex glycerol medium, after which the cells were transferred to buffered complex methanol (0.5%) medium supplemented with 5 µM all-*trans*-retinal (Millipore-Sigma, Cat. #116-31-4) and grown at 30°C with shaking at 230 rpm. After 24 hr, the yellow-colored cells were harvested by centrifugation at 5000 × *g* for 10 min, and the cell pellets were resuspended in 100 ml ice-cold buffer A (20 mM HEPES, pH 7.4, 150 mM NaCl, 1 mM EDTA, 5% glycerol) and lysed by either French press or bead beater. After centrifugation at low speed (5000 × *g* for 10 min) to remove cell debris, membrane fractions were pelleted at 190,000 × *g* for 1 hr using a Ti45 Beckman rotor. The membranes were suspended in Buffer B (350 mM NaCl, 5% glycerol, 20 mM HEPES, pH 7.5) with 1 mM phenylmethylsulfonyl fluoride and solubilized with 1% n-dodecyl-β-D-maltoside (DDM; Anatrace, Cat. # D310) for 1 hr at 4°C with shaking. Undissolved content was removed after ultracentrifugation using a Ti45 rotor at 110,000 × *g* for 1 hr. The supernatant supplemented with 15 mM imidazole was incubated with nickel-nitrilotriacetic acid resin (Qiagen, Cat. # 30210) for 1 hr with shaking at 4°C. The resin was washed step-wise using 15 mM and 40 mM imidazole in Buffer B supplemented with 0.03% DDM. The protein was eluted with 400 mM imidazole and 0.03% DDM in buffer B. Protein fractions were pooled and concentrated using a 50 kDa MWCO Amicon Ultra Centrifugal Filter (Millipore-Sigma, Cat. # UFC9050), flash-frozen in liquid nitrogen and stored at −80°C until use.

## Absorption spectroscopy and flash photolysis

Absorption spectra of detergent-purified protein samples were recorded using a Cary 4000 spectrophotometer (Varian). Photoinduced absorption changes were measured with a laboratory-constructed crossbeam apparatus. Excitation flashes were provided by a Minilite II Nd:YAG laser (532 nm, pulse width 6 ns, energy 5 mJ; Continuum). Measuring light was from a 250 W incandescent tungsten lamp and a McPherson monochromator (model 272, Acton). Absorption changes were detected with a Hamamatsu Photonics photomultiplier tube (model R928) combined with a second monochromator of the same type. Signals were amplified by a low noise current amplifier (model SR445A; Stanford Research Systems) and digitized with a GaGe Octopus digitizer board (model CS8327, DynamicSignals LLC), with a maximal sampling rate of 50 MHz. Logarithmic data filtration was performed using the GageCon program (*Sineshchekov et al., 2023*).

## Mice

All procedures to maintain and use mice were approved by the Institutional Animal Care and Use Committee at Baylor College of Medicine (protocol AN-6544). Mice were maintained on a 14 hr:10 hr light:dark cycle with regular mouse chow and water ad libitum. The temperature was maintained at 21–25°C and humidity at 40–60%. Experiments were performed during the light cycle. Female ICR (CD-1) mice were purchased from Baylor College of Medicine Center for Comparative Medicine, and male C57BL/6 J (JAX #000664) mice were obtained from Jackson Laboratory. Both male and female mice were used in the experiments.

## In utero electroporation

Female ICR mice were crossed with male C57BL/6 J mice to obtain timed pregnancies. In utero electroporation was used to deliver the transgenes (*Xue et al., 2014*). To express *Ans*ACR or *Ft*ACR in the layer 2/3 pyramidal neurons of the somatosensory cortex, pAAV-CAG-*Ans*ACR-EYFP or pAAV-CAG-*Ft*ACR-EYFP (2.5 µg µl⁻¹ final concentration) was mixed with pCAG-tdTomato (0.1 µg µl⁻¹ final concentration) and Fast Green (Sigma-Aldrich, 0.01% final concentration) for injection. On embryonic day 15, pregnant mice were anesthetized, and a beveled glass micropipette (tip size 100 µm outer diameter, 50 µm inner diameter) was used to penetrate the uterus and the embryo skull to inject ~1.5 µl DNA solution into one lateral ventricle. Five pulses of current (voltage 39 V, duration 50ms) were delivered

at 1 Hz with a Tweezertrode (5 mm diameter) and a square-wave pulse generator (Gemini X2, BTX Harvard Bioscience). The electrode paddles were positioned in parallel with the brain's sagittal plane. The cathode contacted the side of the brain ipsilateral to the injected ventricle to target the somatosensory cortex. Transfected pups were identified by the transcranial fluorescence of tdTomato with an MZ10F stereomicroscope (Leica) 1 day after birth.

## Brain slice electrophysiology and imaging

Mice were used at the age of 4–6 weeks for acute brain slice electrophysiology experiments. Mice were anesthetized by an intraperitoneal injection of a ketamine and xylazine mix (80 mg kg$^{-1}$ and 16 mg kg$^{-1}$, respectively) and perfused transcardially with cold (0–4°C) slice cutting solution containing 80 mM NaCl, 2.5 mM KCl, 1.3 mM NaH$_2$PO$_4$, 26 mM NaHCO$_3$, 4 mM MgCl$_2$, 0.5 mM CaCl$_2$, 20 mM $d$-glucose, 75 mM sucrose and 0.5 mM sodium ascorbate (315 mOsm l$^{-1}$, pH 7.4, saturated with 95% O$_2$/5% CO$_2$). Brains were removed and sectioned in the cutting solution with a VT1200S vibratome (Leica) to obtain 300 μm coronal slices. Slices were incubated in a custom-made interface holding chamber containing slice cutting solution saturated with 95% O$_2$/5% CO$_2$ at 34°C for 30 min and then at room temperature for 20 min to 10 hr until they were transferred to the recording chamber. We performed recordings on submerged slices in artificial cerebrospinal fluid (ACSF) containing 119 mM NaCl, 2.5 mM KCl, 1.3 mM NaH$_2$PO$_4$, 26 mM NaHCO$_3$, 1.3 mM MgCl$_2$, 2.5 mM CaCl$_2$, 20 mM $d$-glucose and 0.5 mM sodium ascorbate (305 mOsm l$^{-1}$, pH 7.4, saturated with 95% O$_2$/5% CO$_2$, perfused at 3 ml min$^{-1}$) at 30–32°C. For whole-cell recordings, a K$^+$-based pipette solution containing 142 mM K$^+$ gluconate, 10 mM HEPES, 1 mM EGTA, 2.5 mM MgCl$_2$, 4 mM ATP-Mg, 0.3 mM GTP-Na, 10 mM Na$_2$-phosphocreatine (295 mOsm l$^{-1}$, pH 7.35) was used. Membrane potentials reported in *Figure 7*, *Figure 7—figure supplement 1* were not corrected for LJP, which was 12.5 mV as measured experimentally. Neurons were visualized with video-assisted IR differential interference contrast imaging, and fluorescent neurons were identified by epifluorescence imaging under a water immersion objective (×40, 0.8 NA) on an upright SliceScope Pro 1000 microscope (Scientifica) with an IR-1000 CCD camera (DAGE-MTI). Data were acquired at 10 kHz and low-pass filtered at 4 kHz with an Axon Multiclamp 700B amplifier and an Axon Digidata 1440 A Data Acquisition System under the control of Clampex 10.7 (Molecular Devices). Data were analyzed offline using Clampfit (Molecular Devices). For photostimulation, blue light was emitted from a collimated 470 nm light-emitting diode (LED; M470L3, Thorlabs) to stimulate *Ans*ACR- or *Ft*ACR-expressing neurons. The LEDs were driven by a LED driver (Thorlabs LEDD1B) under the control of an Axon Digidata 1440 A Data Acquisition System and Clampex 10.7. The light was delivered through the reflected light fluorescence illuminator port and the ×40 objective.

To evaluate the inhibition efficiency, action potentials of *Ans*ACR- or *Ft*ACR-expressing neurons were evoked by injecting a series of 1.5 s depolarizing current pulses (−0.1–0.5 nA) in whole-cell current clamp mode. 1 s 470 nm (38.7 mW mm$^{-2}$) light stimulation was applied in the middle of current injections with 30 s inter-trial interval. Resting membrane potentials, input resistances, and capacitances were measured in the trials with –0.1 nA current injection. To examine the excitatory effect of the ACRs, 10 ms 470 nm (38.7 mW mm$^{-2}$) light stimulation was applied, and ACR$^-$ neurons were clamped at –70 mV to record excitatory post-synaptic currents.

After electrophysiology recordings, fluorescent images of the brain slices were acquired on an Axio Zoom.V16 Fluorescence Stereo Zoom Microscope (Zeiss) and processed using MATLAB2024b (MathWorks). Images were taken from 20 brain slices of two male and two female mice.

## Generation of transgenic *C. elegans* strains and EPG recording

The transgenic *C. elegans* strains COP2831 and [pNU3704 ([uncp-17::AnsACR::EYFP::tbb-2u, unc-119(+))]] II; unc-119(ed3) III expressing *Ans*ACR in cholinergic neurons were created by InVivo Biosystems using the Mos1-mediated Single Copy Insertion (MosSCI) method, which enables integrating the transgene as a single-copy insertion at a designated locus in the *C. elegans* genome (*Frøkjær-Jensen, 2015*). *Unc*-119 rescue cassette insertion was used to bring the transgene into a Mos1 target locus on chromosome II and create rescue of the function of the *unc-119(ed3)* III mutant allele. The Mos1 locus was selected for position-neutral effects and to avoid the gene coding regions, introns, and transcription factor binding sites. The integration of the transgene was confirmed by PCR.

The transgenic and Bristol N2 wild-type worms were grown at 20°C on *E. coli* strain OP50 lawns in the absence or presence of 10 µM (final concentration) all-*trans*-retinal (Millipore-Sigma, Cat. # 116-31-4), which was mixed with the bacteria before seeding Nematode Growth Medium (NGM) plates. EPG recordings were performed from intact worms sucked into a pipette (*Raizen and Avery, 1994*). The pipettes (200 kΩ resistance) were pulled from borosilicate glass and filled with the External Physiological solution, the composition of which is specified in the above section. The worms were transferred to the same solution supplemented with 10 mM serotonin before the measurements. The data were acquired in the voltage clamp mode of the same Axopatch 200B amplifier used for manual patch clamp recording from HEK293 cells and the same software. The data obtained in two independently created strains were pooled together. The photoexcitation was provided by the Polychrome V light source described above. The frequency of the R1-spikes was calculated using the Event Detection by the Threshold Search function of ClampFit after applying a 2 Hz high-pass digital filter. Further analysis was performed using the Origin Pro 2016 software.

## Reproducibility and statistics

Plasmids encoding different ChR variants were randomly assigned to transfect identical cell batches. Three independent transfections were performed on different experimental days; the data obtained were pooled together. In automated patch clamp studies, cells were blindly selected by the machine and randomly drawn into the wells. For an unbiased estimation of the photocurrent amplitude, the data from wells that formed seals with a resistance <500 MΩ were excluded. For a more accurate estimation of the $V_r$ values by plotting the IV curves, wells with a seal resistance <500 MΩ and photocurrents of the absolute magnitude <50 pA at –80 mV were excluded. In manual patch clamp experiments, the cells were selected for patching by inspecting their tag fluorescence; non-fluorescent cells and cells in which no GΩ seal was established or lost during recording were excluded from the analysis. Recordings with the access resistance ($R_a$) >20 MΩ were excluded from the analysis. In automated and manual patch clamp experiments, the photocurrent traces recorded from different cells transfected with the same construct were considered biological replicates (reported as n values). These values indicate how often the experiments were performed independently. In experiments using continuous light pulses, only one photocurrent trace was recorded from one cell for each condition. To increase the signal-to-noise ratio for computer approximations of the photocurrent traces under single-turnover conditions, six replicates recorded from the same cells were considered technical replicates and averaged for further analysis.

Statistical analysis of the patch clamp data was performed using Origin Pro 2016 software. The normal distribution of the data was not assumed. The non-parametric two-tailed Mann-Whitney and Kolmogorov-Smirnov tests were used to compare the means. No statistical methods were used to pre-determine sample sizes, but the sample sizes were similar to those reported in the previous publications (*Govorunova et al., 2022a*; *Morizumi et al., 2023*).

In the cortical neuron patch clamp experiments, fluorescently labeled or negative neurons were randomly selected from the densely labeled area of brain slices. The criteria for data exclusion were the same as in manual patch clamp recordings from HEK cells. Statistical analysis of the data was performed using ClampFit 10.7 (Molecular Devices) and Prism 10.3 (GraphPad). The multiple Wilcoxon matched-pairs signed rank test with Benjamini, Krieger, and Yekutieli's corrections was used for evaluating the efficiency of action potential inhibition.

In *C. elegans* experiments, the EPG recordings were excluded from the analysis if the worm moved out of the illuminated area during recording. No data was excluded in flash photolysis experiments.

## Acknowledgements

We thank Dr. Valeria Vasquez (UTHealth) for a generous donation of the wild-type *C elegans* strain and OP50 *E coli* strain. We thank Dr. Edward S Boyden (Massachusetts Institute of Technology) for a gift of the Chrimson plasmid. This work was supported by the National Institutes of Health grants R35GM140838 (J.L.S.), S10OD032293 (J.L.S.), U01NS118288 (M.X., J.L.S., F.S.P.), RF1NS133657 (J.L.S., F.S.P., M.X.), R61CA278458 (F.S.P.), and R01NS136027 (F.S.P.); the Robert A Welch Foundation Endowed Chair AU-0009 (J.L.S.), and grants Q-2016–20220331 (F.S.P.) and Q-2016–20190330 (F.S.P.); a Vivian L Smith Endowed Professorship in Neuroscience (F.S.P); the McNair Medical Foundation (F.S.P); the Natural Sciences and Engineering Research Council of Canada (NSERC) grants

RGPIN-2018–04397 and RGPIN-2024–03857 (L.S.B.). An NSERC USRA award supported A.P., and S.M. was supported by the President's Research Assistantship (PRA) program at the University of Guelph. M.X. is a Caroline DeLuca Scholar.

## Additional information

### Competing interests

Mingshan Xue: was a consultant to Capsida Biotherapeutics. Capsida Biotherapeutics provided research funds to Baylor College of Medicine to support a research project in his lab that is unrelated to this study and had no role in the research, authorship, and publication of this article. The other authors declare that no competing interests exist.

### Funding

| Funder | Grant reference number | Author |
|---|---|---|
| National Institutes of Health | R35GM140838 | John L Spudich |
| National Institutes of Health | S10OD032293 | John L Spudich |
| National Institutes of Health | U01NS118288 | François St-Pierre Mingshan Xue John L Spudich |
| National Institutes of Health | RF1NS133657 | François St-Pierre Mingshan Xue John L Spudich |
| National Institutes of Health | R61CA278458 | François St-Pierre |
| National Institutes of Health | R01NS136027 | François St-Pierre |
| Welch Foundation | AU-0009 | John L Spudich |
| Welch Foundation | Q-2016-20220331 | François St-Pierre |
| Welch Foundation | Q-2016-20190330 | François St-Pierre |
| Vivian L. Smith Foundation | | François St-Pierre |
| Natural Sciences and Engineering Research Council of Canada | RGPIN-2018-04397 | Leonid S Brown |
| Natural Sciences and Engineering Research Council of Canada | RGPIN-2024-03857 | Leonid S Brown |
| University of Guelph | President's Research Assistantship | Stephen Mitchell |
| Natural Sciences and Engineering Research Council of Canada | USRA award | Alyssa Palmateer |

The funders had no role in study design, data collection and interpretation, or the decision to submit the work for publication.

### Author contributions

Elena G Govorunova, Conceptualization, Data curation, Formal analysis, Investigation, Visualization, Methodology, Writing – original draft, Project administration, Writing – review and editing; Oleg A Sineshchekov, Conceptualization, Data curation, Formal analysis, Investigation, Visualization, Methodology, Writing – original draft, Writing – review and editing; Hai Li, Investigation, Visualization, Methodology, Writing – review and editing; Yueyang Gou, Data curation, Investigation, Visualization,

Methodology, Writing – original draft, Writing – review and editing; Hongmei Chen, Investigation; Shuyuan Yang, Data curation, Formal analysis, Investigation, Visualization, Methodology, Writing – original draft, Writing – review and editing; Yumei Wang, Investigation, Methodology, Writing – review and editing; Stephen Mitchell, Formal analysis, Investigation, Writing – review and editing; Alyssa Palmateer, Data curation, Investigation, Writing – review and editing; Leonid S Brown, Conceptualization, Resources, Data curation, Formal analysis, Supervision, Funding acquisition, Investigation, Methodology, Writing – original draft, Project administration, Writing – review and editing; François St-Pierre, Mingshan Xue, Conceptualization, Resources, Data curation, Supervision, Funding acquisition, Methodology, Writing – original draft, Project administration, Writing – review and editing; John L Spudich, Conceptualization, Resources, Supervision, Funding acquisition, Project administration, Writing – review and editing

Author ORCIDs
Elena G Govorunova ⓘ https://orcid.org/0000-0003-0522-9683
Hai Li ⓘ https://orcid.org/0000-0002-3969-6709
François St-Pierre ⓘ https://orcid.org/0000-0001-8618-4135
Mingshan Xue ⓘ https://orcid.org/0000-0003-1463-8884
John L Spudich ⓘ https://orcid.org/0000-0003-4167-8590

Ethics
All procedures to maintain and use mice were approved by the Institutional Animal Care and Use Committee at Baylor College of Medicine (protocol AN-6544).

Reviewer #1 (Public review): https://doi.org/10.7554/eLife.106508.3.sa1
Reviewer #2 (Public review): https://doi.org/10.7554/eLife.106508.3.sa2
Reviewer #3 (Public review): https://doi.org/10.7554/eLife.106508.3.sa3
Author response https://doi.org/10.7554/eLife.106508.3.sa4

## Additional files

### Supplementary files
Supplementary file 1. Compositions and liquid junction potential (LJP) values of the solutions used in automated patch clamp recording.

Supplementary file 2. The wavelength positions of the half-maximal amplitude of the long-wavelength slope of the spectrum ($\lambda_{50}$) of ChR variants tested in this study.

MDAR checklist

### Data availability
The numerical data and statistical analyses are provided in the Source Data Files. The sequence information was deposited at the NCBI with GenBank Acc. #PQ657777-PQ657783. The plasmids encoding AnsACR, FtACR, and NlCCR expression constructs in the pcDNA3.1 vector backbone were deposited at Addgene (plasmids #232598, 232599, and 232600, respectively). The plasmids pAAV-CAG-AnsACR-EYFP and pAAV-CAG-FtACR-EYFP were deposited at Addgene (plasmids #238347 and #238348, respectively). The recombinant *C. elegans* lines expressing AnsACR in the cholinergic neurons are available from the authors upon request.

The following datasets were generated:

| Author(s) | Year | Dataset title | Dataset URL | Database and Identifier |
|---|---|---|---|---|
| Govorunova EG, Sineshchekov OA, Li H, Gou Y, Chen H, Yang S, Wang Y, Mitchell S, Palmateer A, Brown LS, St-Pierre F, Xue M, Spudich JL | 2025 | Synthetic construct clone AnsACR anion channelrhodopsin gene, partial cds | https://www.ncbi.nlm.nih.gov/nuccore/PQ657777 | NCBI GenBank, PQ657777 |

*Continued*

| Author(s) | Year | Dataset title | Dataset URL | Database and Identifier |
|---|---|---|---|---|
| Govorunova EG, Sineshchekov OA, Li H, Gou Y, Chen H, Yang S, Wang Y, Mitchell S, Palmateer A, Brown LS, St-Pierre F, Xue M, Spudich JL | 2025 | Synthetic construct clone ParsR non-electrogenic channel rhodopsin-like protein gene, partial cds | https://www.ncbi. nlm.nih.gov/nuccore/ PQ657783 | NCBI GenBank, PQ657783 |
| Govorunova EG, Sineshchekov OA, Li H, Gou Y, Chen H, Yang S, Wang Y, Mitchell S, Palmateer A, Brown LS, St-Pierre F, Xue M, Spudich JL | 2025 | Synthetic construct clone FtACR anion channel rhodopsin gene, partial cds | https://www.ncbi. nlm.nih.gov/nuccore/ PQ657778.1/ | NCBI GenBank, PQ657778 |
| Govorunova EG, Sineshchekov OA, Li H, Gou Y, Chen H, Yang S, Wang Y, Mitchell S, Palmateer A, Brown LS, St-Pierre F, Xue M, Spudich JL | 2025 | Synthetic construct clone NlCCR channel rhodopsin gene, partial cds | https://www.ncbi. nlm.nih.gov/nuccore/ PQ657779 | NCBI GenBank, PQ657779 |
| Govorunova EG, Sineshchekov OA, Li H, Gou Y, Chen H, Yang S, Wang Y, Mitchell S, Palmateer A, Brown LS, St-Pierre F, Xue M, Spudich JL | 2025 | Synthetic construct clone AtACR anion channel rhodopsin gene, partial cds | https://www.ncbi. nlm.nih.gov/nuccore/ PQ657780.1/ | NCBI GenBank, PQ657780 |
| Govorunova EG, Sineshchekov OA, Li H, Gou Y, Chen H, Yang S, Wang Y, Mitchell S, Palmateer A, Brown LS, St-Pierre F, Xue M, Spudich JL | 2025 | Synthetic construct clone OaACR1 anion channel rhodopsin gene, partial cds | https://www.ncbi. nlm.nih.gov/nuccore/ PQ657781 | NCBI GenBank, PQ657781 |
| Govorunova EG, Sineshchekov OA, Li H, Gou Y, Chen H, Yang S, Wang Y, Mitchell S, Palmateer A, Brown LS, St-Pierre F, Xue M, Spudich JL | 2025 | Synthetic construct clone OaACR2 anion channel rhodopsin gene, partial cds | https://www.ncbi. nlm.nih.gov/nuccore/ PQ657782 | NCBI GenBank, PQ657782 |

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
