## [Editor Report · eLife Assessment]

This **important** study describes newly identified light-gated ion channel homologs (channelrhodopsins, ChRs) in several protist species, with a primary focus on the biophysical characterization of ChRs of ancyromonads. The authors employed a powerful combination of bioinformatics, manual and automated patch-clamp electrophysiology, absorption spectroscopy, and flash photolysis. Additionally, they evaluated the applicability of the newly discovered anion-conducting ChRs in cortical neurons of mouse brain slices and in living *C. elegans* worms. The evidence supporting most of the claims is **compelling**, and this work will be of interest to the microbial rhodopsin community and neuro- and cardioscientists utilizing optogenetics in their research.

---

## [Referee Report · Reviewer #1 (Public review)]

Summary:

This work by Govorunova et al. identified three naturally blue-shifted channelrhodopsins (ChRs) from ancyromonads, namely AnsACR, FtACR, and NlCCR. The phylogenetic analysis places the ancyromonad ChRs in a distinct branch, highlighting their unique evolutionary origin and potential for novel applications in optogenetics. Further characterization revealed the spectral sensitivity, ionic selectivity, and kinetics of the newly discovered AnsACR, FtACR, and NlCCR. This study also offers valuable insights into the molecular mechanism underlying the function of these ChRs, including the roles of specific residues in the retinal-binding pocket. Finally, this study validated the functionality of these ChRs in both mouse brain slices (for AnsACR and FtACR) and in vivo in *Caenorhabditis elegans* (for AnsACR), demonstrating the versatility of these tools across different experimental systems.

In summary, this work provides a potentially valuable addition to the optogenetic toolkit by identifying and characterizing novel blue-shifted ChRs with unique properties.

Strengths:

This study provides a thorough characterization of the biophysical properties of the ChRs' properties and demonstrated the versatility of these tools in different ex vivo and in vivo experimental systems. The authors also explored the potential of AnsACR for multiplexed optogenetics. Finally, the mutagenesis experiments revealed the roles of key residues in the photoactive site that can affect the spectral and kinetic properties of the channelrhodopsins.

Weaknesses:

The revised manuscript has addressed most of the previous major weaknesses.

---

## [Referee Report · Reviewer #2 (Public review)]

Summary:

Govorunova et al present three new anion opsins that have potential applications silencing neurons. They identify new opsins by scanning numerous databases for sequence homology to known opsins, focusing on anion opsins. The three opsin identified, are uncommonly fast, potent, and are able to silence neuronal activity. The authors characterize numerous parameters of the opsins and compare these opsins to the existing and widely used GtACR opsins.

Strengths:

This paper follows the tradition of the Spudich lab, presenting and rigorously characterizing potentially valuable opsins. Furthermore, they explore several mutations of the identified opsin that may make these opsins even more useful for the broader community. The opsins AnsACR and FtACR are particularly notable having extraordinarily fast onset kinetics that could have utility in many domains. Furthermore, the authors show AnsACR is useable in multiphoton experiments having a peak photocurrent in a commonly used wavelength. Overall, the author's detailed measurements and characterization make for an important resource - both presenting new opsins that may be important for future experiment, and providing characterizations to expand our understanding of opsin biophysics in general.

---

## [Referee Report · Reviewer #3 (Public review)]

Summary:

The authors aimed to develop Channelrhodopsins (ChRs), light-gated ion channels, with high potency and blue action spectra for use in multicolor (multiplex) optogenetics applications. To achieve this, they performed a bioinformatics analysis to identify ChR homologues in several protist species, focusing on ChRs from ancyromonads, which exhibited the highest photocurrents and the most blue-shifted action spectra among the tested candidates. Within the ancyromonad clade, the authors identified two new anion-conducting ChRs and one cation-conducting ChR. These were characterized in detail using a combination of manual and automated patch-clamp electrophysiology, absorption spectroscopy, and flash photolysis. The authors also explored sequence features that may explain the blue-shifted action spectra and differences in ion selectivity among closely related ChRs.

Strengths:

A key strength of this study is the high-quality experimental data, which were obtained using well-established techniques such as manual patch-clamp and absorption spectroscopy, complemented by modern automated patch-clamp approaches. These data convincingly support most of the claims. The newly characterized ChRs expand the optogenetics toolkit and will be of significant interest to researchers working with microbial rhodopsins, those developing new optogenetic tools, as well as neuro- and cardioscientists employing optogenetic methods.

Weaknesses:

This study does not exhibit major methodological weaknesses.

---

## [Author Response]

**Reviewer #1 (Public review):**
Summary:This work by Govorunova et al. identified three naturally blue-shifted channelrhodopsins (ChRs) from ancyromonads, namely AnsACR, FtACR, and NlCCR. The phylogenetic analysis places the ancyromonad ChRs in a distinct branch, highlighting their unique evolutionary origin and potential for novel applications in optogenetics. Further characterization revealed the spectral sensitivity, ionic selectivity, and kinetics of the newly discovered AnsACR, FtACR, and NlCCR. This study also offers valuable insights into the molecular mechanism underlying the function of these ChRs, including the roles of specific residues in the retinal-binding pocket. Finally, this study validated the functionality of these ChRs in both mouse brain slices (for AnsACR and FtACR) and in vivo in *Caenorhabditis elegans* (for AnsACR), demonstrating the versatility of these tools across different experimental systems.In summary, this work provides a potentially valuable addition to the optogenetic toolkit by identifying and characterizing novel blue-shifted ChRs with unique properties.Strengths:This study provides a thorough characterization of the biophysical properties of the ChRs and demonstrates the versatility of these tools in different ex vivo and in vivo experimental systems. The mutagenesis experiments also revealed the roles of key residues in the photoactive site that can affect the spectral and kinetic properties of the channel.

We thank the Reviewer for his/her positive evaluation of our work.

Weaknesses:While the novel ChRs identified in this work are spectrally blue-shifted, there still seems to be some spectral overlap with other optogenetic tools. The authors should provide more evidence to support the claim that they can be used for multiplex optogenetics and help potential end-users assess if they can be used together with other commonly applied ChRs. Additionally, further engineering or combination with other tools may be required to achieve truly orthogonal control in multiplexed experiments.

To demonstrate the usefulness of ancyromonad ChRs for multiplex optogenetics as a proof of principle, we co-expressed AnsACR with the red-shifted cation-conducting ChR Chrimson and measured net photocurrent generated by this combination as a function of the wavelength. We found that it is hyperpolarizing in the blue region of the spectrum, and depolarizing at the red region. In the revision, we added a new panel (Figure 1D) showing these results and the following paragraph to the main text:

“To test the possibility of using AnsACR in multiplex optogenetics, we co-expressed it with the red-shifted CCR Chrimson (Klapoetke et al., 2014) fused to an EYFP tag in HEK293 cells. We measured the action spectrum of the net photocurrents with 4 mM Cl^-^ in the pipette, matching the conditions in the neuronal cytoplasm (Doyon, Vinay et al. 2016). Figure 1D, black shows that the direction of photocurrents was hyperpolarizing upon illumination with λ<500 nm and depolarizing at longer wavelengths. A shoulder near 520 nm revealed a FRET contribution from EYFP (Govorunova, Sineshchekov et al. 2020), which was also observed upon expression of the Chrimson construct alone (Figure 1D, red)”.

In the *C. elegans* experiments, partial recovery of pharyngeal pumping was observed after prolonged illumination, indicating potential adaptation. This suggests that the effectiveness of these ChRs may be limited by cellular adaptation mechanisms, which could be a drawback in long-term experiments. A thorough discussion of this challenge in the application of optogenetics tools would prove very valuable to the readership.

We added the following paragraph to the revised Discussion:

“One possible explanation of the partial recovery of pharyngeal pumping that we observed after 15-s illumination, even at the highest tested irradiance, is continued attenuation of photocurrent during prolonged illumination (desensitization). However, the rate of AnsACR desensitization (Figure 1 – figure supplement 4A and Figure 1 – figure supplement 5A) is much faster than the rate of the pumping recovery, reducing the likelihood that desensitization is driving this phenomenon. Another possible reason for the observed adaptation is an increase in the cytoplasmic Cl^-^ concentration owing to AnsACR activity and hence a breakdown of the Cl^-^ gradient on the neuronal membrane. The *C. elegans* pharynx is innervated by 20 neurons, 10 of which are cholinergic (Pereira, Kratsios et al. 2015). A pair of MC neurons is the most important for regulation of pharyngeal pumping, but other pharyngeal cholinergic neurons, including I1, M2, and M4, also play a role (Trojanowski, Padovan-Merhar et al. 2014). Moreover, the pharyngeal muscles generate autonomous contractions in the presence of acetylcholine tonically released from the pharyngeal neurons (Trojanowski, Raizen et al. 2016). Given this complexity, further elucidation of pharyngeal pumping adaptation mechanisms is beyond the scope of this study.”

**Reviewer #2 (Public review):**
Summary:Govorunova et al present three new anion opsins that have potential applications in silencing neurons. They identify new opsins by scanning numerous databases for sequence homology to known opsins, focusing on anion opsins. The three opsins identified are uncommonly fast, potent, and are able to silence neuronal activity. The authors characterize numerous parameters of the opsins.Strengths:This paper follows the tradition of the Spudich lab, presenting and rigorously characterizing potentially valuable opsins. Furthermore, they explore several mutations of the identified opsin that may make these opsins even more useful for the broader community. The opsins AnsACR and FtACR are particularly notable, having extraordinarily fast onset kinetics that could have utility in many domains. Furthermore, the authors show that AnsACR is usable in multiphoton experiments having a peak photocurrent in a commonly used wavelength. Overall, the author's detailed measurements and characterization make for an important resource, both presenting new opsins that may be important for future experiments, and providing characterizations to expand our understanding of opsin biophysics in general.

We thank the Reviewer for his/her positive evaluation of our work.

Weaknesses:First, while the authors frequently reference GtACR1, a well-used anion opsin, there is no side-by-side data comparing these new opsins to the existing state-of-the-art. Such comparisons are very useful to adopt new opsins.

GtACR1 exhibits the peak sensitivity at 515 nm and therefore is poorly suited for combination with red-shifted CCRs or fluorescent sensors, unlike blue-light-absorbing ancyromonad ACRs. Nevertheless, we conducted side-by-side comparison of ancyromonad ChRs, GtACR1 and GtACR2, the latter of which has the spectral maximum at 470 nm. The results are shown in the new Figures 1E and F, and the new multipanel Figure 1 – figure supplement 4 added in the revision. We also added the following text, describing these results, to the revised Results section:

“Figures 1E and F show the dependence of the peak photocurrent amplitude and reciprocal peak time, respectively, on the photon flux density for ancyromonad ChRs and GtACRs. The current amplitude saturated earlier than the time-to-peak for all tested ChRs. Figure 1 – figure supplement 4A-E shows normalized photocurrent traces recorded at different photon densities. Quantitation of desensitization at the end of 1-s illumination revealed a complex light dependence (Figure 1, Figure Supplement 4F). Figure 1 – figure supplement 5 shows normalized photocurrent traces recorded in response to a 5-s light pulse of the maximal available intensity and the magnitude of desensitization at its end.”

Next, multiphoton optogenetics is a promising emerging field in neuroscience, and I appreciate that the authors began to evaluate this approach with these opsins. However, a few additional comparisons are needed to establish the user viability of this approach, principally the photocurrent evoked using the 2p process, for given power densities. Comparison across the presented opsins and GtACR1 would allow readers to asses if these opsins are meaningfully activated by 2P.

We carried out additional 2P experiments in ancyromonad ChRs, GtACR1 and GtACR2 and added their results to a new main-text Figure 6 and Figure 6 – figure supplement 1. We added the new section describing these results, “Two-photon excitation”, to the main text in the revision:

“To determine the 2P activation range of AnsACR, FtACR, and NlCCR, we conducted raster scanning using a conventional 2P laser, varying the excitation wavelength between 800 and 1,080 nm (Figure 6 – figure supplement 1). All three ChRs generated detectable photocurrents with action spectra showing maximal responses at ~925 nm for AnsACR, 945 nm for FtACR, and 890 nm for NlCCR (Figure 6A). These wavelengths fall within the excitation range of common Ti:Sapphire lasers, which are widely used in neuroscience laboratories and can be tuned between ~700 nm and 1,020-1,300 nm. To assess desensitization, cells expressing AnsACR, FtACR, or NlCCR were illuminated at the respective peak wavelength of each ChR at 15 mW for 5 seconds. GtACR1 and GtACR2, previously used in 2P experiments (Forli, Vecchia et al. 2018, Mardinly, Oldenburg et al. 2018), were included for comparison. The normalized photocurrent traces recorded under these conditions are shown in Figure 6B-F. The absolute amplitudes of 2P photocurrents at the peak time and at the end of illumination are shown in Figure 6G and H, respectively. All five tested variants exhibited comparable levels of desensitization at the end of illumination (Figure 6I).”

**Reviewer #3 (Public review):**
Summary:The authors aimed to develop Channelrhodopsins (ChRs), light-gated ion channels, with high potency and blue action spectra for use in multicolor (multiplex) optogenetics applications. To achieve this, they performed a bioinformatics analysis to identify ChR homologues in several protist species, focusing on ChRs from ancyromonads, which exhibited the highest photocurrents and the most blue-shifted action spectra among the tested candidates. Within the ancyromonad clade, the authors identified two new anion-conducting ChRs and one cation-conducting ChR. These were characterized in detail using a combination of manual and automated patch-clamp electrophysiology, absorption spectroscopy, and flash photolysis. The authors also explored sequence features that may explain the blue-shifted action spectra and differences in ion selectivity among closely related ChRs.Strengths:A key strength of this study is the high-quality experimental data, which were obtained using well-established techniques such as manual patch-clamp and absorption spectroscopy, complemented by modern automated patch-clamp approaches. These data convincingly support most of the claims. The newly characterized ChRs expand the optogenetics toolkit and will be of significant interest to researchers working with microbial rhodopsins, those developing new optogenetic tools, as well as neuro- and cardioscientists employing optogenetic methods.

We thank the Reviewer for his/her positive evaluation of our work.

Weaknesses:This study does not exhibit major methodological weaknesses. The primary limitation of the study is that it includes only a limited number of comparisons to known ChRs, which makes it difficult to assess whether these newly discovered tools offer significant advantages over currently available options.

We conducted side-by-side comparison of ancyromonad ChRs and GtACRs, wildly used for optical inhibition of neuronal activity. The results are shown in the new Figures 1E and F, and the new multipanel Figure 1 – figure supplement 4 and Figure 1 – figure supplement 5 added in the revision. We also added the following text, describing these results, to the revised Results section:

“Figures 1E and F show the dependence of the peak photocurrent amplitude and reciprocal peak time, respectively, on the photon flux density for ancyromonad ChRs and GtACRs. The current amplitude saturated earlier than the time-to-peak for all tested ChRs. Figure 1 – figure supplement 4A-E shows normalized photocurrent traces recorded at different photon densities. Quantitation of desensitization at the end of 1-s illumination revealed a complex light dependence (Figure 1, Figure Supplement 4F). Figure 1 – figure supplement 5 shows normalized photocurrent traces recorded in response to a 5-s light pulse of the maximal available intensity and the magnitude of desensitization at its end.”

Additionally, although the study aims to present ChRs suitable for multiplex optogenetics, the new ChRs were not tested in combination with other tools. A key requirement for multiplexed applications is not just spectral separation of the blue-shifted ChR from the red-shifted tool of interest but also sufficient sensitivity and potency under low blue-light conditions to avoid cross-activation of the respective red-shifted tool. Future work directly comparing these new ChRs with existing tools in optogenetic applications and further evaluating their multiplexing potential would help clarify their impact.

As a proof of principle, we co-expressed AnsACR with the red-shifted cation-conducting CCR Chrimson and demonstrated that the net photocurrent generated by this combination is hyperpolarizing in the blue region of the spectrum, and depolarizing at the red region. In the revision, we added a new panel (Figure 1D) showing these results and the following paragraph to the main text:

“To test the possibility of using AnsACR in multiplex optogenetics, we co-expressed it with the red-shifted CCR Chrimson (Klapoetke et al., 2014) fused to an EYFP tag in HEK293 cells. We measured the action spectrum of the net photocurrents with 4 mM Cl^-^ in the pipette, matching the conditions in the neuronal cytoplasm (Doyon, Vinay et al. 2016). Figure 1D, black shows that the direction of photocurrents was hyperpolarizing upon illumination with λ<500 nm and depolarizing at longer wavelengths. A shoulder near 520 nm revealed a FRET contribution from EYFP (Govorunova, Sineshchekov et al. 2020), which was also observed upon expression of the Chrimson construct alone (Figure 1D, red)”.

**Reviewing Editor Comments:**
The reviewers suggest that direct comparison to GtACR1 is the most important step to make this work more useful to the community.

We followed the Reviewers’ recommendations and carried out side-by-side comparison of ancyromonad ChRs and GtACR1 as well as GtACR2 (Figure 1E and F, Figure 1 – figure supplement 4, Figure 1 – figure supplement 5, and Figure 6). Note, however, that GtACR1’s spectral maximum is at 515 nm, which makes it poorly suitable for blue light excitation. Also, ChRs are known to perform very differently in different cell types and upon expression of their genes in different vector backbones, so our results cannot be generalized for all experimental systems. Each ChR user needs to select the most appropriate tool for his/her purpose by testing several candidates in his/her own experimental setting.

**Reviewer #1 (Recommendations for the authors):**
(1) The figure legend for Figure 2D-I appears to be incomplete. Please provide a detailed explanation of the panels.

In the revision, we have expanded the legend of Figure 2 to explain all individual panels.

(2) The meaning of the Vr shift (Y-axis in Figure 2H-I) should be clarified in the main text to aid reader understanding.

In the revision, we added the phrase “which indicated higher relative permeability to NO_3_ than to Cl^-“^ to explain the meaning of the Vr shift upon replacement of Cl^-^ with NO_3_-.

(3) Adding statistical analysis for the peak and end photocurrent values in Figure 2D-F would strengthen the claim that there is minimal change in relative permeability during illumination.

In the revision, we added the V_r_ values for the peak photocurrent to Figure 2H-I, which already contained the V_r_ values for the end photocurrent, and carried out a statistical analysis of their comparison. The following sentence was added to the text in the revision:

“The V_r_ values of the peak current and that at the end of illumination were not significantly different by the two-tailed Wilcoxon signed-rank test (Fig. 2G), indicating no change in the relative permeability during illumination.”

(4) Figure 4H and I seem out of place in Figure 4, as the title suggests a focus on wild-proteins and AnsACR mutants. The authors could consider moving these panels to Figure 3 for better alignment with the content.

As noted below, we changed the panel order in Figure 4 upon the Reviewer’s request. In particular, former Figure 4I is Figure 4C in the revision, and former Figure 4H is now panel C in Figure 3 – figure supplement 1 in the revision. We rearranged the corresponding section of the text (highlighted yellow in the manuscript).

(5) The characterization section could be strengthened by including data on the pH sensitivity of FtACR, which is currently missing from the main figures.

Upon the Reviewer’s request, we carried out pH titration of FtACR absorbance and added the results as Figure 4B in the revision.

(6) The logic in Figure 4A-G appears somewhat disjointed. For example, Figure 4A shows pH sensitivity for WT AnsACR and the G86E mutant, while Figure 4 B-D shifts to WT AnsACR and the D226N mutant, and Figure 4E returns to the G86E mutant. Reorganizing or clarifying the flow would improve readability.

We followed the Reviewer’s advice and changed the panel order in Figure 4. In the revised version, the upper row (panels A-C) shows the pH titration data of the three WTs, the middle row (panels D-F) shows analysis of the AnsACR_D226N mutant, and the lower row (panels G-I) shows analysis of the AnsACR_G88E mutant. We also rearranged accordingly the description of these panels in the text.

(7) In Figure 5A, "NIACR" should likely be corrected to "NlCCR".

We corrected the typo in the revision.

(8) The statistical significance in Figure 6C and D is somewhat confusing. Clarifying which groups are being compared and using consistent symbols would improve interoperability.

In the revision, we improved the figure panels and legend to clarify that the comparisons are between the dark and light stimulation groups within the same current injection.

(9) The authors pointed out that at rest or when a small negative current was injected, the neurons expressing Cl- permeable ChRs could generate a single action potential at the beginning of photostimulation, as has been reported before. The authors could help by further discussing if and how this phenomenon would affect the applicability of such tools.

We mentioned in the revised Discussion section that activation of ACRs in the axons could depolarize the axons and trigger synaptic transmission at the onset of light stimulation, and this undesired excitatory effect need to be taken into consideration when using ACRs.

**Reviewer #2 (Recommendations for the authors):**
Govorunova et al present three new anion opsins that have potential applications in silencing neurons. This paper follows the tradition of the Spudich lab, presenting and rigorously characterizing potentially valuable opsins. Furthermore, they explore several mutations of the identified opsin that may make these opsins even more useful for the broader community. In general, I feel positively about this manuscript. It presents new potentially useful opsins and provides characterization that would enable its use. I have a few recommendations below, mostly centered around side-by-side comparisons to existing opsins.(1) My primary concern is that while there is a reference to GtACR1, a highly used opsin first described by this team, they do not present any of this data side by side.When evaluating opsins to use, it is important to compare them to the existing state of the art. As a potential user, I need to know where these opsins differ. Citing other papers does not solve this as, even within the same lab, subtle methodological differences or data plotting decisions can obscure important differences.

As we explained in the response to the public comments, we carried out side-by-side comparison of ancyromonad ChRs and GtACRs as requested by the Reviewer. The results are shown in the new Figures 1E and F, and the new multipanel Figure 1 – figure supplement 4 and Figure 1 – figure supplement 5, added in the revision. However, we would like to emphasize a limited usefulness of such comparative analysis, as ChRs are known to perform very differently in different cell types and upon expression of their genes in different vector backbones, so our results cannot be generalized for all experimental systems. Each ChR user needs to select the most appropriate tool for his/her purpose by testing several candidates in his/her own experimental setting.

(2) Multiphoton optogenetics is an emerging field of optogenetics, and it is admirable that the authors address it here. The authors should present more 2p characterization, so that it can be established if these new opsins are viable for use with 2P methods, the way GtACR1 is. The following would be very useful for 2P characterization:Photocurrents for a given power density, compared to GtACR1 and GtACR2.

The new Figure 6 (B-F) added in the revision shows photocurrent traces recorded from the three ancyromonad ChRs and two GtACRs upon 2P excitation of a given power density.

Comparing NICCR and FtACR's wavelength specificity and photocurrent. If these opsins are too weak to create reasonable 2P spectra, this difference should be discussed.

The new Figure 6A shows the 2P action spectra of all three ancyromonad ChRs.

A Trace and calculated photocurrent kinetics to compare 1P and 2P. This need not be the flash-based absorption characterization of Figure 3, but a side-by-side photocurrent as in Figure 2.

As mentioned above, photocurrent traces recorded from ancyromonad ChRs and GtACRs upon 2P excitation are shown in the new Figure 6 (B-F). However, direct comparison of the 2P data with the 1P data is not possible, as we used laser scanning illumination for the former and wild-field illumination for the latter.

Characterization of desensitization. As the authors mention, many opsins undergo desensitization, presenting the ratio of peak photocurrent vs that at multiple time points (probably up to a few seconds) would provide evidence for how effectively these constructs could be used in different scenarios.We conducted a detailed analysis of desensitization under both 1P and 2P excitation. The new Figure 1 – figure supplement 4 and Figure 1 – figure supplement 5 show the data obtained under 1P excitation, and the new Figure 6 shows the data for 2P conditions.I have to admit, that by the end of the paper, I was getting confused as to which of the three original constructs had which property, and how that was changing with each mutation. I would suggest that a table summarizing each opsin and mutation with its onset and offset kinetics, peak wavelength, photocurrent, and ion selectivity would greatly increase the ability to select and use opsins in the future.

In the revision, we added a table of the spectroscopic properties of all tested mutants as Supplementary File 2. This study did not aim to analyze other parameters listed by the Reviewer. We added the following sentence referring to this table to the main text:

“Supplementary File 2 contains the λ values of the half-maximal amplitude of the long-wavelength slope of the spectrum, which can be estimated more accurately from the action spectra than the λ of the maximum.”

It may be out of the scope of this manuscript, but if a soma localization sequence can be shown to remove the 'axonal spiking' (as described in line 441), this would be a significant addition to the paper.

Our previous study (Messier et al., 2018, doi: 10.7554/eLife.38506) showed that a soma localization sequence can reduce, but not eliminate, the axonal spiking. We plan to test these new ACRs with the trafficking motifs in the future.

NICCR appears to have the best photocurrents of all tested opsins in this paper. It seems odd that it was omitted from the mouse cortical neurons experiments.

We have not included analysis of NlCCR behavior in neurons because we are preparing a separate manuscript on this ChR.

Figure 6 would benefit from more gradation in the light powers used to silence and would benefit from comparison to GtACR. I suggest using a fixed current with a series of illumination intensities to see which of the three opsins (or GtACR) is most effective at silencing. At present, it looks binary, and a user cannot evaluate if any of these opsins would be better than what is already available.

In the revision, we added the data comparing the light sensitivity of AnsACR and FtACR with previously identified GtACR1 and GtACR2 (new Figure 1E and F) to help users compare these ACRs. Although they are less sensitive to light comparing to GtACR1 and GtACR2, they could still be activated by commercially available light sources if the expression levels are similar. Less sensitive ACRs may have less unwanted activation when using with other optogenetic tools.

**Reviewer #3 (Recommendations for the authors):**
Suggested Improvements to Experiments, Data, or Analyses:(1) Line 25: "significantly exceeding those by previously known tools" and Line 408: "NlCCR is the most blue-shifted among ancyromonad ChRs and generates larger photocurrents than the earlier known CCRs with a similar absorption maximum." As noted in the public review, this statement applies only to a very specific subgroup of ChRs with spectral maxima below 450 nm. If the goal was to claim that NlCCR is a superior tool among a broader range of blue-light-activated ChRs, direct comparisons with state-of-the-art ChRs such as ChR2 T159C (Berndt et al., 2011), CatCh (Kleinlogel et al., 2014), CoChR (Klapoetke et al., 2014), CoChR-3M (Ganjawala et al., 2019), or XXM 2.0 (Ding et al., 2022) would be beneficial. If the goal was to demonstrate superiority among tools with spectra below 450 nm, I suggest explicitly stating this in the paper.

The Reviewer correctly inferred that we emphasized the superiority of NlCCR among tools with similar spectral maxima, not all blue-light-activated ChRs available for neuronal photoexcitation, most of which exhibit absorption maxima at longer wavelengths. To clarify this, we added “with similar spectral maxima” to the sentence in the original Line 25. The sentence in Line 408 already contains this clarification: “with a similar absorption maximum”.

(2) Lines 111-113: "The absorption spectra of the purified proteins were slightly blue-shifted from the respective photocurrent action spectra (Figure 1D), likely due to the presence of non-electrogenic cis-retinal-bound forms." I would be skeptical of this statement. The spectral shifts in NlCCR and AnsACR are small and may fall within the range of experimental error. The shift in FtACR is more apparent; however, if two forms coexist in purified protein, this should be reflected as two Gaussian peaks in the absorption spectrum (or at least as a broader total peak reflecting two states with close maxima and similar populations). On the contrary, the action spectrum appears to have two peaks, one potentially below 465 nm. Generally, neither spectrum appears significantly broader than a typical microbial rhodopsin spectrum. This question could be clarified by quantifying the widths of the absorption and action spectra or by overlaying them on the same axis. In my opinion, the two spectra seem very similar, and just appearance of the "bump" in the action spectum shifts the apparent maximum of the action spectrum to the red. If there were two states, then they should both be electrogenic, and the slight difference in spectra might be explained by something else (e.g. by a slight difference in the quantum yields of the two states).

As the Reviewer suggested, in the revision we added a new figure (Figure 1 – figure supplement 2), showing the overlay of the absorption and action spectra of each ancyromonad ChR. This figure shows that the absorption spectra are wider than the action spectra (especially in AnsACR and FtACR), which confirms our interpretation (contribution of the non-electrogenic blue-shifted cis-retinal-bound forms to the absorption spectrum). Note that the presence of such forms explaining a blue shift of the absorption spectrum has been experimentally verified in HcKCR1 (doi: 10.1016/j.cell.2023.08.009; 10.1038/s41467-025-56491-9). Therefore, we revised the text as follows:

“The absorption spectra of the purified proteins (Figure 1C) were slightly blue-shifted from the respective photocurrent action spectra (Figure 1 – figure supplement 3), likely due to the presence of non-electrogenic cis-retinal-bound forms. The presence of such forms, explaining the discrepancy between the absorption and the action spectra, was verified by HPLC in KCRs (Tajima et al. 2023, Morizumi et al., 2025).”

(3) Lines 135-136: "The SyncroPatch enables unbiased estimation of the photocurrent amplitude because the cells are drawn into the wells without considering their tag fluorescence." While SyncroPatch does allow unbiased selection of patched cells, it does not account for the fraction of transfected cells. Without a method to exclude non-transfected cells, which are always present in transient transfections, the comparison of photocurrents may be affected by the proportion of untransfected cells, which could vary between constructs. To clarify whether the statistically significant difference in the Kolmogorov-Smirnov test could indicate that the fraction of transfected cells after 48-72h differs between constructs, I suggest analyzing only transfected cells or reporting fractions of transfected cells by each construct.

The Reviewer correctly states that non-transfected cells are always present in transiently transfected cell populations. However, his/her suggestion to “exclude non-transfected cells” is not feasible in the absence of a criterion for such exclusion. As it is evident from our data, transient transfection results in a continuum of the amplitude values, and it is not possible to distinguish a small photocurrent from no photocurrent, considering the noise level. We would like, however, to emphasize that not excluding any cells provides an estimate of the overall potency of each ChR variant, which depends on both the fraction of transfected cells and their photocurrents. This approach mimics the conditions of in vivo experiments, when non-expressing cells also cannot be excluded.

(4) Line 176: "AnsACR and FtACR photocurrents exhibited biphasic rise." The fastest characteristic time is very close to the typical resolution of a patch-clamp experiment (RC = 50 μs for a 10 pF cell with a 5 MΩ series resistance). Thus, I am skeptical that the faster time constant of the biphasic opening represents a protein-specific characteristic time. It may not be fully resolved by patch-clamp and could simply result from low-pass filtering of a specific cell. I suggest clarifying this for the reader.

The Reviewer is right that the patch clamp setup acts as a lowpass filter. Earlier, we directly measured its time resolution (~15 μs) by recording the ultrafast (occurring on the ps time scale) charge movements related to the trans-cis isomerization (doi: 10.1111/php.12558). However, the lowpass filter of the setup can only slow the entire signal, but cannot lead to the appearance of a separate kinetic component (i.e. a monophasic process cannot become biphasic). Therefore, we believe that the biphasic photocurrent rise reflects biphasic channel opening rather than a measurement artifact. Two phases in the channel opening have also been detected in GtACR1 (doi: 10.1073/pnas.1513602112) and CrChR2 (10.1073/pnas.1818707116).

(5) Line 516: "The forward LED current was 900 mA." It would be more informative to report the light intensity rather than the forward current, as many readers may not be familiar with the specific light output of the used LED modules at this forward current.

We have added the light intensity value in the revision:

“The forward LED current was 900 mA (which corresponded to the irradiance of ~2 mW mm^-2^)…”

(6) Lines 402-403: "The NlCCR ... contains a neutral residue in the counterion position (Asp85 in BR), which is typical of all ACRs. Yet, NlCCR does not conduct anions, instead showing permeability to Na+." This is not atypical for CCRs and has been demonstrated in previous works of the authors (CtCCR in Govorunova et al. 2021, ChvCCR1 in Govorunova et al. 2022). What is unique is the absence of negatively charged residues in TM2, as noted later in the current study. However, the absence of negatively charged residues in TM2 appears to be rare for ACRs as well. Not as a strong point of criticism, but to enhance clarity, I suggest analyzing the frequency of carboxylate residues in TM2 of ACRs to determine whether the unique finding is relevant to ion selectivity or to another property.

The Reviewer is correct that some CCRs lack a carboxylate residue in the D85 position, so this feature alone cannot be considered as a differentiating criterion. However, the complete absence of glutamates in TM2 is not rare in ACRs and is found, for example, in HfACR1 and CarACR2. We have discussed this issue in our earlier review (doi: 10.3389/fncel.2021.800313) and do not think that repeating this discussion in this manuscript is appropriate.

**Recommendations for Writing and Presentation:**
(1) Some figures contain incomplete or missing labels:Figure 2: Panels D to I lack labels.

In the revision, we have expanded the legend of Figure 2 to explain all individual panels.

Figure 3 - Figure Supplement 1: Missing explanations for each panel.

In the revision, we changed the order of panes and explained all individual panels in the legend.

Figure 5 - Figure Supplement 1: Missing explanations for each panel.

No further explanation for individual panels in this Figure is needed because all panels show the action spectra of various mutants, the names of which are provided in the panels themselves. Repeating this information in the figure legend would be redundant.

(2) In Figure 2, "sem" is written in lowercase, whereas "SEM" is capitalized in other figures. Standardizing the format would improve consistency.

In the revision, we changed the font of the SEM abbreviation to the uppercase in all instances.

(3) Line 20: "spectrally separated molecules must be found in nature." There is no proof that they cannot be developed synthetically; rather, it is just difficult. I suggest softening this statement, as the findings of this study, together with others, will probably allow designing molecules with specified spectral properties in the future.

In the revision, we changed the cited sentence to the following:

“Multiplex optogenetic applications require spectrally separated molecules, which are difficult to engineer without disrupting channel function”.

(4) Line 216-219: "Acidification increased the amplitude of the fast current ~10-fold (Figure 4F) and shifted its Vr ~100 mV (Figure 3 - figure supplement 1D), as expected of passive proton transport. The number of charges transferred during the fast peak current was >2,000 times smaller than during the channel opening, from which we concluded that the fast current reflects the movement of the RSB proton." The claim about passive transport of the RSB proton should be clarified, as typically, passive transport is not limited to exactly one proton per photocycle, and the authors observe the increase in the fast photocurrents upon acidification.

We thank the Reviewer for pointing out the confusing character of our description. To clarify the matter, we added a new photocurrent trace to Figure 4I in the revision recorded from AnsACR_G86E at 0 mV and pH 7.4. We have rewritten the corresponding section of Results as follows:

“Its rise and decay τ corresponded to the rise and decay τ of the fast positive current recorded from AnsACR_G86E at 0 mV and neutral pH, superimposed on the fast negative current reflecting the chromophore isomerization (Figure 4I, upper black trace). We interpret this positive current as an intramolecular proton transfer to the mutagenetically introduced primary acceptor (Glu86), which was suppressed by negative voltage (Figure 4I, lower black trace). Acidification increased the amplitude of the fast negative current ~10-fold (Figure 4I, black arrow) and shifted its V_r_ ~100 mV to more depolarized values (Figure 4 – figure supplement 2A). This can be explained by passive inward movement of the RSB proton along the large electrochemical gradient.”

Minor Corrections:(1) Line 204: Missing bracket in "phases in the WT (Figure 4D)."

The quoted sentence was deleted during the revision.

(2) Line 288: Typo-"This Ala is conserved" should probably be "This Met is conserved."

We mean here the Ala four residues downstream from the first Ala. To avoid confusion, we changed the cited sentence to the following:

“The Ala corresponding to BR’s Gly122 is also found in AnsACR and NlCCR (Figure 5A)…”

(3) Lines 702-704: Missing Addgene plasmid IDs in "(plasmids #XXX and #YYY, respectively)."

In the revision, we added the missing plasmid IDs.